# A lead-width distribution for Antarctic sea ice: a case study for the Weddell Sea with high resolution Sentinel-2 images

Marek Muchow[1], Amelie U. Schmitt[2], and Lars Kaleschke[3]

[1]Institute of Oceanography, Center for Earth System Research and Sustainability (CEN), Universität Hamburg, Hamburg, GERMANY
[2]Meteorological Institute, Center for Earth System Research and Sustainability (CEN), Universität Hamburg, Hamburg, GERMANY
[3]Alfred-Wegener-Institut, Helmholtz-Zentrum für Polar- und Meeresforschung, Bremerhaven, GERMANY

**Correspondence:** Marek Muchow (marek.muchow@uni-hamburg.de)

**Abstract.** Using Copernicus Sentinel-2 images we derive a statistical lead-width distribution for the Weddell Sea. While previous work focused on the Arctic, this is the first lead-width distribution for Antarctic sea ice. Previous studies suggest that the lead-width distribution follows a power law with a positive exponent, however their results for the power-law exponents are not all in agreement with each other.

To detect leads we create a sea-ice surface-type classification based on 20 carefully selected cloud-free Sentinel-2 Level 1C products, which have a resolution of 10 m. The observed time period is from November 2016 until February 2018, covering only the months from November to April. We apply two different fitting methods to the measured lead widths. The first fitting method is a linear fit, while the second method is based on a maximum likelihood approach. Here, we use both methods for the same lead-width data set to observe differences in the calculated power-law exponent.

To further investigate influences on the power-law exponent, we define two different thresholds one for open water and one for open water and nilas covered leads. The influence of the lead threshold on the exponent is larger for the linear fit than for the method based on the maximum likelihood approach. We show that the exponent of the lead-width distribution ranges between 1.110 and 1.413 depending on the applied fitting method and lead threshold. This exponent for the Weddell Sea sea ice is smaller than the previously observed exponents for the Arctic sea ice.

*Copyright statement.* TEXT

## 1 Introduction

Leads are created by dynamic motions of the sea ice (Miles and Barry, 1998) and covered by open water or thin sea ice. They often follow a linear-like shape, can be up to tens of kilometers long and are by definition a few meters to some kilometers wide (e.g. Alam and Curry, 1997). An adequate representation of leads in climate models is important for various processes.

Leads play a large role in the absorption of shortwave radiation due to the low albedo of open water and nilas, compared to the

higher albedo of thicker ice and snow covered sea ice (Perovich, 1996). Newly formed leads are also an important area for ice production and the associated brine rejection to the ocean below (Alam and Curry, 1997).

Furthermore, the heat exchange between atmosphere and ocean is strongly enhanced over leads. Using a simple heat flux model, Maykut (1978) found that the heat loss over thin ice (0.4 - 0.5 m) is one magnitude larger than over multiyear ice. In a model study, Lüpkes et al. (2008) demonstrated that an increase in the lead fraction area by 1 % during polar night can lead to local air temperature warming of up to 3.5 K. Based on buoy data in the Weddell Sea region combined with a thermodynamic sea ice model, Eisen and Kottmeier (2000) found that leads contribute roughly 30 % to the total energy flux from the ocean to the atmosphere in winter months. Due to the large temperature differences between the air and the lead surface in winter, convective plumes forming over leads can have a large impact on the atmospheric processes in regions covered with sea ice (e.g. Tetzlaff et al., 2015; Lüpkes et al., 2008; Chechin et al., 2019).

Different studies suggested that the overall heat exchange over leads does not only depend on lead area fraction or ice thickness, but also on lead width. Using a fetch-dependent formulation of the heat exchange, Marcq and Weiss (2012) demonstrated that the heat transfer is two times more effective for narrow leads of several meters than for wider ones of several hundreds of meters. Furthermore, Qu et al. (2019) used a combination of remote sensing and reanalysis data and found that narrow leads ($\leq 1$ km) accounted for about a quarter of the heat flux over all leads.

To account for these lead-width-dependent processes in models the lead width needs to be parametrized. One possibility is to apply a lead-width distribution. Several studies estimating shear and divergence rates for Arctic sea ice using satellite observations suggest that these quantities follow a power law (e.g. Marsan et al., 2004; Stern and Lindsay, 2009). Such a power law scaling has also been found in different modeling studies (e.g. Girard et al., 2009; Wang et al., 2016; Ólason et al., 2021). Since leads are formed by divergent sea ice motions, it is plausible to also expect a power law behaviour for lead width. Power-law exponents for lead widths in the Arctic have been derived from submarine measurements (Wadhams, 1981; Wadhams et al., 1985), as well as remote sensing data from thermal imagers (Lindsay and Rothrock, 1995; Qu et al., 2019), visible imagery (Marcq and Weiss, 2012), and altimetry (Wernecke and Kaleschke, 2015). Since data with different resolutions were used in these studies, there are substantial differences in the methods used to detect leads and of the minimum considered lead widths. In addition, different statistical methods have been applied to calculate the power-law exponents. Consequently, obtained values for the power-law exponent from observations vary in absolute values and suitable range of the distribution.

For the Antarctic, different studies have derived lead fractions (Allison et al., 1993; Reiser et al., 2020; Petty et al., 2021), however lead width distributions have not been studied, yet. In this study, we derive a lead-width distribution for the Weddell Sea sea ice as a case study for Antarctic sea ice. For this purpose, we introduce a new method to derive lead widths using Sentinel-2 data. The main goals of this study are 1) to demonstrate that Sentinel-2 data are suitable for deriving lead widths and 2) to determine whether a power law behavior - with an exponent similar to previous results for the Arctic - can also be found for Antarctic sea ice in the Weddell Sea.

The main advantage of the recently launched Sentinel-2 satellites is their high resolution up to 10 m. This enables us to detect also very narrow leads, which most of the former studies were not capable of. We use cloud-free Sentinel-2 Level-1C products, which give the top-of-the-atmosphere (TOA) reflectance (Drusch et al., 2012). The data are described in Section 2. Similar to

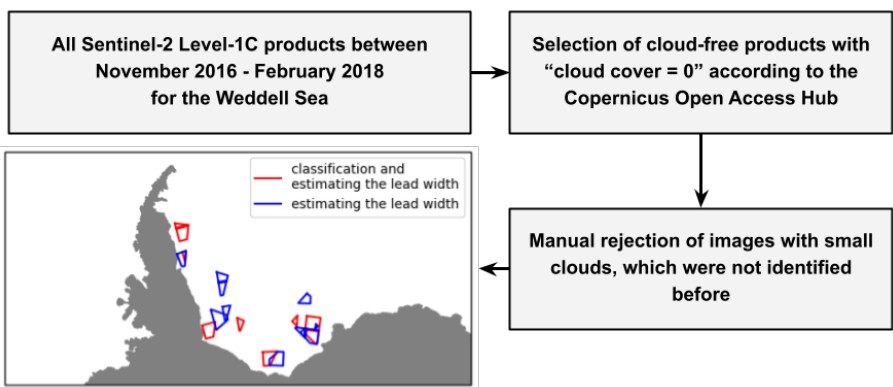

**Figure 1.** Display of the selection steps for the 20 Sentinel-2 Level-1C products. The location of the 20 different Sentinel-2 Level-1C products for this study is the Weddell Sea. Nine out of the 20 were used for the sea ice surface-type classification (red border), while for the lead-width detection all 20 were used (red and blue border). For the border of the product the "real image outlines" are displayed, which are not always rectangular since the satellite swath does not always overlap completely with the processing grid applied by ESA. Displayed in gray is Antarctic continent border including shelf ice border measured with different satellite radar from 2007 - 2009 (Mouginot et al., 2017; Rignot et al., 2013)

the albedo for young, thin sea ice the TOA reflectance is related to the ice thickness. As a first step, we introduce a surface-type classification for the Sentinel-2 satellite products to identify different sea ice types and leads (Sect. 3.1). The determined reflectance thresholds for leads covered with open water and nilas are then used to detect leads and calculate a lead-width distribution. Since some of the previous studies focused on leads covered only by open water and others also included leads covered by thin sea ice, we apply two different reflectance thresholds and compare the results. Subsequently, a power law is fitted to the resulting lead-width distribution. We apply two different statistical methods to determine the power-law exponents, which have both been used in different previous studies, and compare the results (Sect. 3.2). The results are presented and discussed in Section 4, followed by conclusions in Section 5.

## 2 Data

The two sun synchronous Sentinel-2 satellites carry the passively working Multi Spectral Instrument (MSI) with 13 different spectral bands from $443\,\mathrm{nm}$ (visible) to $2190\,\mathrm{nm}$ (short wave infra-red) (ESA, 2018). The spatial resolution for the bands is either 10, 20 or $60\,\mathrm{m}$ while the images cover an area of $100 \times 100\,\mathrm{km}$. A higher resolution allows for the detection of narrower leads. We therefore visually compared all $10\,\mathrm{m}$ bands (2, 3, 4 and 8) to identify the band with the best representation of thin ice structures. The best results were found for band 4 ($665\,\mathrm{nm}$), which is then used for the analysis in this study.

We selected the Weddell Sea as a case study, since Sentinel-2 is a land mission and acquires data over oceans only in the vicinity of land (Drusch et al., 2012) which restricts the regional selection. Due to the need for sunlight to capture suitable data, only Sentinel-2 Level-1C products covering the months from November to April were used. The Weddell Sea contains

**Table 1.** Sentinel-2 Level-1C products used for measuring the lead width. Products which are also used for the classification are labeled with 'yes'.

| Sensing date | Classification | Product name |
|---|---|---|
| 12/11/2016 | no | S2A_MSIL1C_20161112T104212_N0204_R122_T26CMC_20161112T104210 |
| 20/11/2016 | no | S2A_MSIL1C_20161120T100152_N0204_R093_T25CES_20161120T100153 |
| 20/11/2016 | no | S2A_MSIL1C_20161120T100152_N0204_R093_T25CDS_20161120T100153 |
| 29/11/2016 | no | S2A_MSIL1C_20161129T103152_N0204_R079_T24CXE_20161129T103151 |
| 20/12/2016 | no | S2A_MSIL1C_20161220T100052_N0204_R093_T24CVV_20161220T100049 |
| 23/02/2017 | yes | S2A_MSIL1C_20170223T123141_N0204_R023_T21CVT_20170223T123144 |
| 23/02/2017 | no | S2A_MSIL1C_20170223T123141_N0204_R023_T22DDF_20170223T123144 |
| 23/02/2017 | no | S2A_MSIL1C_20170223T123141_N0204_R023_T22DDG_20170223T123144 |
| 24/02/2017 | yes | S2A_MSIL1C_20170224T120231_N0204_R037_T22CEC_20170224T120234 |
| 26/02/2017 | yes | S2A_MSIL1C_20170226T110241_N0204_R065_T23CNQ_20170226T110244 |
| 02/03/2017 | no | S2A_MSIL1C_20170302T122211_N0204_R123_T22CDD_20170302T122205 |
| 13/03/2017 | no | S2A_MSIL1C_20170313T101141_N0204_R136_T25CDS_20170313T101144 |
| 16/03/2017 | yes | S2A_MSIL1C_20170316T102141_N0204_R036_T25CES_20170316T102141 |
| 16/03/2017 | yes | S2A_MSIL1C_20170316T102141_N0204_R036_T25CES_20170316T102141 |
| 16/03/2017 | yes | S2A_MSIL1C_20170316T102141_N0204_R036_T24CWC_20170316T102141 |
| 06/04/2017 | yes | S2A_MSIL1C_20170406T131051_N0204_R052_T21DVF_20170406T131050 |
| 06/04/2017 | yes | S2A_MSIL1C_20170406T131051_N0204_R052_T21DVG_20170406T131050 |
| 06/04/2017 | yes | S2A_MSIL1C_20170406T131051_N0204_R052_T21DVD_20170406T131050 |
| 06/04/2017 | no | S2A_MSIL1C_20170406T131051_N0204_R052_T20DPJ_20170406T131050 |
| 09/02/2018 | no | S2A_MSIL1C_20180209T120241_N0206_R037_T21CWU_20180209T163245 |

a large enough sea-ice cover during these months (e.g. Comiso and Nishio, 2008). Additionally, only products classified as cloud-free were selected in the Copernicus Open Access Hub (https://scihub.copernicus.eu/dhus/#/home). We noticed that on products with wide leads often small clouds occur, most likely from moisture and heat flux through the lead. Those images were rejected manually and we only use totally cloud-free images. Thus, the final 20 Sentinel-2 Level-1C products are always between the months of November to April, while the whole observation period ranges from November 2016 until February 2018 (Figure 1).

The lead-width detection method (Sect. 3.2) is applied to all 20 products. The classification of surface types and threshold identification (Sect. 3.1) is based on 9 of those 20 products from January to April 2017. For more details on the data see table

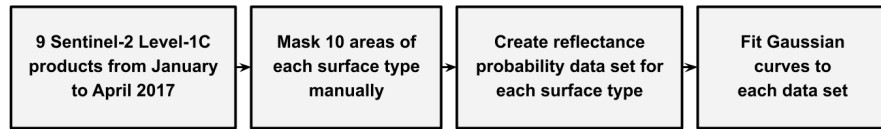

**Figure 2.** Data analysis steps for obtaining the Gaussian curves for each surface type.

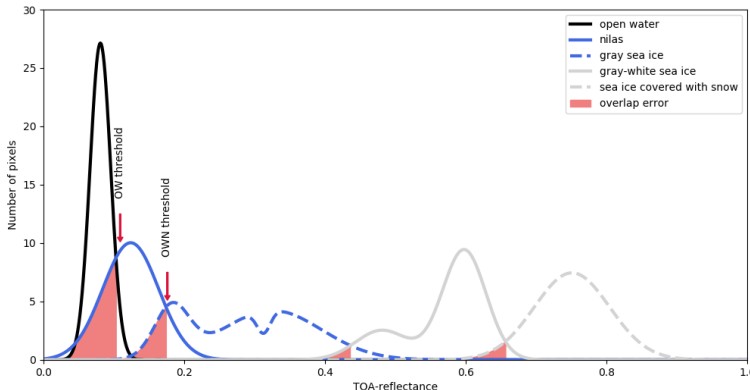

**Figure 3.** The number of pixels within every surface type for a specific TOA reflectance. The TOA reflectance threshold for each surface type is the point of intersection of two curves adjacent to each other. The error is shown as the overlap error of these two curves below each threshold. The red arrows show the two thresholds later used for the lead identification for the lead-width measurement: the open water (OW) threshold and the open water and nilas (OWN) threshold.

1.

# 3 Methods

## 3.1 Threshold identification

The threshold identification contains the following main steps (Figure 2): First, the classification of five different surface types based on the top-of-the-atmosphere (TOA) reflectance. Second, the creation of a TOA reflectance probability data set for each surface type and the fit of Gaussian curves to each data set. Third, the results from the surface classification are used to identify two thresholds, which are later used for creating binary "lead-sea ice"-images for the lead-width measurement.

For the surface-type classification nine out of 20 later used Sentinel-2 Level-1C products are utilized (Sect. 2). We identify five different surface types including open water and four different ice types (nilas, gray sea ice, gray-white sea ice and sea ice covered with snow). The names of the sea-ice categories are based on the WMO Sea-ice Nomenclature (WMO, 2014) for

consistency with other literature. However, we want to stress that our classification is based on the TOA reflectance and not on sea-ice age or thickness. On every band 4 image ten areas of each surface type are masked manually. Thereafter, the TOA reflectance of each pixel within the mask is used to create a reflectance value data set for each surface type. The reflectance values lie between zero and one.

To analyze the range of the TOA reflectance for each surface type, histograms are created, which show the occurrence of pixels with a specific TOA reflectance. These histograms are used to fit a summation over Gaussian functions with the mean $\mu$ and standard deviation $\sigma$ to the data:

$$y(x) = \sum_{i=1}^{n} a_i \cdot \frac{1}{\sqrt{2\pi}\sigma_i} \cdot e^{-0.5(\frac{x-\mu_i}{\sigma_i})^2} \tag{1}$$

$n$ indicates the number of Gaussian curves, that were combined to one function and weighted with the weighting parameter
$a_i$, for fitting the histograms. By using $n > 1$ we can account for multiple maxima in a distribution. Thus, $n = 2$ is used for gray-white sea ice and with $n = 3$ for gray sea ice (Figure 3). One Gaussian curve ($n = 1$) is fitted to the histogram for open water, nilas and sea ice covered with snow.

The threshold for each surface category are then determined as the values of the TOA reflectance at the point of intersection of two curves adjacent to each other. An exception is the threshold for open water, where two points of intersection occur. In this
case the second point of intersection is chosen to be the threshold, because the first point of intersection is before the maximum. The area of intersection of two curves is then the overlap error of those thresholds and describes where we manually classified pixels with the same TOA-reflectance in different sea-ice surface categories.

For the lead identification two different thresholds are used to create binary images: one for leads covered with open water (OW threshold) and one for leads covered with open water and nilas (OWN threshold). We decided to use two thresholds to observe
the effect of the coverage of the lead on the power law similar to Marcq and Weiss (2012), who used two different luminance thresholds for leads. Additionally, we decided to use the combined OWN threshold since open water refreezes quickly in leads depending on the surrounding temperatures, but the leads keep similar properties in regards to head exchange as open water leads. Additionally, leads are defined as being navigable by surface vessels (WMO, 2014), which is still true for leads covered with nilas.


## 3.2 Measuring the apparent lead width and determining the power-law exponent

Since the leads within each image can have arbitrary orientations, it is not guaranteed to measure the "true lead width" orthogonally to the leads orientation, but the width of a line across the lead at an angle other then $90°$. As in Wernecke and Kaleschke (2015) we call the then measured lead width the apparent lead width as a proxy for the true lead width. To measure the apparent
lead width we use a measurement grid consisting of ten vertical and ten horizontal equally spaced measurement tracks across each Sentinel-2 product (Figure 4).

The obtained data set of apparent lead widths can then be displayed as a histogram showing the occurrence $p(x)$ for each specific width. As has been done in previous studies (Wadhams, 1981; Wadhams et al., 1985; Lindsay and Rothrock, 1995;

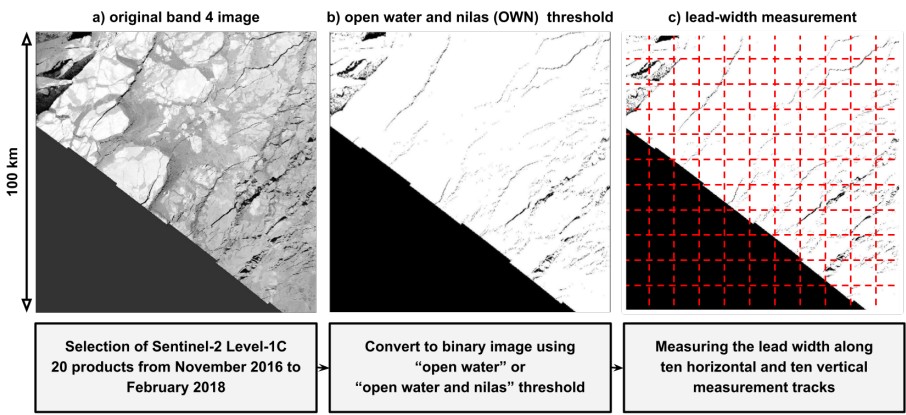

**Figure 4.** a) Exemplary original Sentinel-2 Level-1C band 4 image (sensing date: 16/03/2017). b) Binary image after the application of the open water and nilas (OWN) threshold, where leads are indicated with black pixels and no leads with white ones. c) Applied measurement grid with ten horizontal and vertical measurement tracks. The swath of the Sentinel-2 satellite does not cover the whole image area defined by the ESA data-processing grid. Thus, only the area covered by the satellite swath is considered for the lead-width measurement.

Marcq and Weiss, 2012; Wernecke and Kaleschke, 2015; Qu et al., 2019), we assume that the shape of the histogram follows a power law with the exponent $\alpha$ and the apparent lead widths $x_{width}$:

$$p(x) = C \cdot x_{width}^{-\alpha} \tag{2}$$

The scaling parameter $C$ is the offset at the y-axis and therefore related to the number of measurements and it is not further investigated here.

We apply two different methods to estimate the power-law exponent $\alpha$. For the linear fit (LF method) the apparent lead widths are sorted by size, so that the frequency $p(x)$ of the specific width is available. On a plot with both logarithmic axes, the distribution of the data follows a straight line with specific slope and an axis intercept. The slope is the representation of the power-law exponent $\alpha$. Due to the same influence of every value for the result of the fit, atypical values have a strong effect on the result (Berk, 2004).

The second method for estimating the exponent $\alpha$ is the method for discrete values by Clauset et al. (2009), which is based on a maximum likelihood approach (ML method). The power law distribution diverges at zero, therefore a lower boundary $x_{min} > 0$ is needed. In this study, $x_{min}$ is the smallest possible apparent lead width, which is the image resolution of 10 m. The following equation is used for estimating the power-law exponent $\alpha$:

$$\alpha \cong 1 + n \cdot \left[ \sum_{i=1}^{n} \ln \left( \frac{x_{width,i}}{x_{min} - \frac{1}{2} \cdot step\ size} \right) \right]^{-1} \tag{3}$$

The total number of counted leads is $n$, and $x_{width,i}$ are the measured lead widths. Since the data are discrete with a resolution of 10 m, the $stepsize$ in equation 3 is set to 10 m similar to Wernecke and Kaleschke (2015).

To reduce the influence of possible single outlining measurements on the result of the power-law exponent, we estimated the

**Table 2.** The table displays the threshold for each surface type from the surface classification. The thresholds are the point of intersection between the Gaussian curves describing the occurred TOA reflectance values for each surface type (Figure 3, Section 3.1). Every threshold contains the surface types, which are above it in the table. Sea ice covered with snow has no estimated threshold, therefore it is indicated as 1.0.

| Surface type | Threshold [TOA reflectance] | Overlap error [%] |
|---|---|---|
| open water | 0.10 | |
| | | 29 |
| nilas | 0.17 | |
| | | 11 |
| dark-gray sea ice | 0.44 | |
| | | 3 |
| light-gray sea ice | 0.66 | |
| | | 4 |
| sea ice covered with snow | 1.0 | |

lead-width distribution one hundred times with a random selection of 70 % of the measured apparent lead widths. We choose 70 % to still have enough measured widths, while having variation between the data sets. The final power-law exponent is then estimated as the mean over the one hundred calculations. Additionally, as a measure for uncertainty, the standard deviation is also estimated from the one hundred calculations.

## 4   Results and discussion

### 4.1   Threshold identification

The thresholds between surface categories and corresponding overlap errors are determined using the method described in Section 3.1. With Sentinel-2 band 4 images it is possible to distinguish between five different surface types (open water, nilas, gray sea ice, gray-white sea ice, sea ice covered with snow) based on top-of-the-atmosphere (TOA) reflectance values (Figure 3). The results for the thresholds and the corresponding overlap error are presented in Table 2. Note that for the lead identification only two thresholds are applied: the open water (OW) threshold and a threshold combining open water and nilas (OWN).

The common value used to compare optical properties of sea ice is the albedo. In this study, we measure TOA reflectance instead of albedo. Both properties increase with the sea ice and snow cover thickness, especially for young, thin sea ice in absence of melting processes. In addition to this, we only use cloud-free Sentinel-2 band 4 images. Thus, the atmosphere has a negligible influence on the reflectance measurement. We estimated the thresholds with Sentinel-2 band 4 images from January to April 2017 to include different sun and look angles. Before estimating the thresholds we also compared the TOA reflectance values for each surface type within the products with each other and found no significant difference. To evaluate the two thresh-

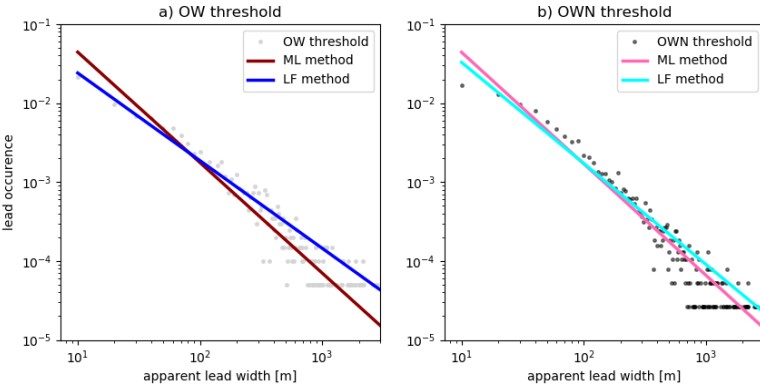

**Figure 5.** Relative lead occurrence as a function of measured lead width (dots). Lead widths were measured using a) the open water (OW) threshold and b) the open water and nilas (OWN) threshold. Straight lines indicate the fitted power law curves using the ML and LF method.

olds, which are later used for the lead detection, they are compared to measured albedo values from the East Antarctic sea ice zone in Australian spring and summer by Brandt et al. (2005). Their estimated albedos for open water (0.07) and nilas without snow cover (0.14) are close to the thresholds estimated here for the same surface types. For the classification of the two later used thresholds we aimed to classify structures without snow cover. For the other surface types it is much more difficult to

make assumptions about the snow cover or thickness, due to the fact that only the reflectance values are known. Nevertheless, our estimated TOA reflectance thresholds for each surface type is always in the range of the reference albedo measurements from Brandt et al. (2005).

Additionally, since leads normally have sharp edges the selection of areas as example values for open water and nilas was comparatively easy compared to the other sea-ice surface types. The thicker the ice and snow cover the more unreliable these

observations become. To obtain a more precise classification of the surface types validation with other data sources like field measurements could be beneficial. Nevertheless, the TOA reflectance thresholds (0.10 for OW and 0.16 for OWN) were used for the lead detection and agree with values from previous measurements (Brandt et al., 2005).

## 4.2 Measured lead widths and the power-law exponent

The lead-width distribution derived from 20 Sentinel-2 products for using both the open water (OW) and the open water and nilas (OWN) threshold is presented in Figure 5. The total number of leads observed with the OW threshold is 2024, while for the OWN threshold 3799 leads are observed. The largest observed apparent lead widths are 6500 m for the OW threshold and 6530 m for the OWN threshold. Looking at the distribution of the measured lead widths it is evident that the small leads dominate and that with an increasing width the number of leads decreases. We measured leads with a width from 10 m down to

the resolution of the Sentinel-2 band 4 image resolution and upwards, but the amount of measured leads with a width of 10 m is less then what might be expected (Figure 5 and 6). One possible reason is the resolution itself and according to Wernecke

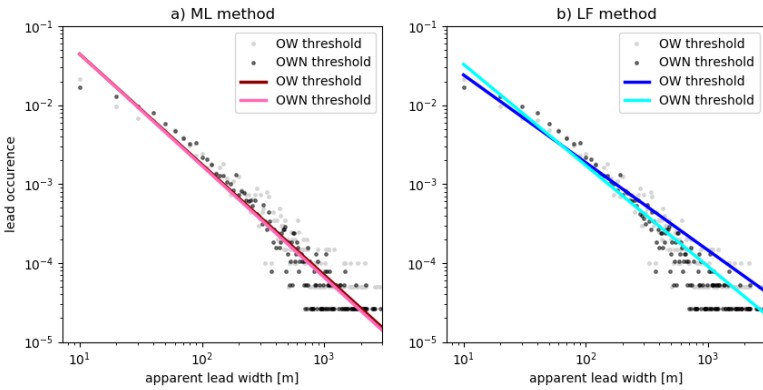

**Figure 6.** Same as Figure 5, but with the results for both thresholds for a) the ML fitting method and b) the LF fitting method.

and Kaleschke (2015) this is a typical feature with less measurements for the lower bound of the resolution, since a 10 m lead is not always covered completely by one image pixel but partially by two or more, so that the signal of the lead is not detected. The upper limit of the power law range is cut off by the availability of wider leads, since wider leads tend to produce small clouds and we only analyzed cloud-free data.

As described in Section 3.2 we apply two different methods to fit a power law to the lead-width distribution. The calculated power-law exponents for both thresholds and fitting methods are presented at the bottom of Table 3. At first we compare the results for the same thresholds with different methods to one another (Figure 5) to estimate the impact of the methods. The values for the power-law exponent with the OW threshold are 1.110 (LF method) and 1.399 (ML method). For this threshold, the the method has a strong implication on the result. For the OWN threshold the results are closer (LF method: 1.280, ML

method: 1.413). The standard deviation for the LF method is ten times higher (0.02) than for the ML method (0.002). These results confirm that the method has a non neglectable effect on the result of the exponent for the sea ice width distribution power law.

Secondly, we compare the results for the same method with both thresholds to show the importance of the choice of thresholds (Figure 6). The OW threshold covers only leads without any thin sea ice, while the OWN threshold includes open water but

also leads covered with sea ice. Thus, the OWN threshold data set includes more lead-width measurements but also wider leads due to lead edges covered with nilas. For the LF method the different thresholds give two different results of the exponent for the lead-width distribution power law (OW: 1.110, OWN: 1.280). Otherwise, for the ML method the choice of the threshold has no strong influence on the result of the power-law exponent (OW: 1.399, OWN: 1.413). Thus, choosing different thresholds or criteria for the definition of the lead can influence the result. This is supported by the result of Marcq and Weiss (2012), who

used two differing thresholds which have a similar range to one another as our estimates for the LF method (Table 3).

Previous studies about lead-width distributions (Table 3) focused on different regions in the Arctic and not on Antarctic regions. While observing leads in the Arctic sea ice is outside the scope of this study, we compare our results with the results from the Arctic sea ice to gain more insight about possible effects on the differences. The exponent of the lead-width distribution power

law determined by in this study for the Weddell Sea sea ice is smaller than in all previous studies for Arctic sea ice: The results by Wernecke and Kaleschke (2015) using the CryoSat-2 satellite support the earlier mentioned results by Marcq and Weiss (2012) (SPOT satellite) with a power-law exponent around 2.50. The power-law exponent found by Qu et al. (2019) (2.241 - 2.346) using a combination of MODIS and Landsat 8 is in the same range as the first and lower exponent from Marcq and Weiss (2012), who also used two thresholds. Furthermore, there were two surveys using submarines from which power-law exponents of 2.00 and 2.29 were calculated (Wadhams (1981) and Wadhams et al. (1985)). The only result below 2.0 is from Lindsay and Rothrock (1995) with a power-law exponent of 1.60. They used data from an Advanced Very High Resolution Radiometer (AVHRR).

In addition to the different measurement systems (different satellites and submarines), different methods regarding to lead definition and measurement, the studies for the Arctic observe leads in different regions (Table 3). Willmes and Heinemann (2016) showed that the sea-ice wintertime lead frequencies differ throughout the Arctic Ocean and identify the marginal ice zone in the Fram Strait and the Barents Sea as the primary region for lead activities. Lead-frequency distributions in the pan-Arctic indicate an influence of bathymetry and ocean currents. However, the result for the lead-width distribution by Lindsay and Rothrock (1995) disagrees also with the result from Marcq and Weiss (2012), which were both obtained in the Central Arctic Ocean, while other previous results are similar (Marcq and Weiss, 2012; Wernecke and Kaleschke, 2015; Qu et al., 2019). Furthermore, the results for the power-law exponent displayed in Table 3 are based on a scale invariant approach, however Qu et al. (2019) used different resolutions of the measured lead width ranging from $30\,\mathrm{m}$ to $1\,\mathrm{km}$ resulting in differences in the power-law exponent in the first decimal place indicating that the power law scaling for lead width might not always be scale invariant. In addition to that, Rampal et al. (2019) confirmed a multi-fractal dependence of the sea ice deformation rates on time and space scales. Thus, applying these results on different processes related to deformation, like leads formed due to divergence, would be a necessary step for further research.

Another possible reason for the differences are the different conditions in both regions. While the Arctic Ocean is surrounded by land mass, the Southern Ocean is surrounding the Antarctic Continent. The Antarctic sea ice is exposed to the Antarctic Circumpolar Current and strong circumpolar winds. The Antarctic sea-ice cover is generally more divergent than much of the Arctic ice cover (Gloersen et al., 1993). Lead fractions in the Central Arctic shown by Petty et al. (2021) are lower compared to the Southern Ocean, which also shows some regional differences. Additionally, Worby et al. (2008) estimated the long-term mean (1981 - 2005) of total Antarctic sea ice thickness in winter as $0.66 \pm 0.60\,\mathrm{m}$. For the Arctic Ocean, Kwok et al. (2009) calculated a 5-year mean (2003 - 2008) ice thickness during winter of $2.9 \pm 0.3\,\mathrm{m}$. Different sea ice thicknesses influence the sea ice to have different rheologic properties (Feltham, 2008).

## 5  Conclusions

We introduce a lead-width distribution for Antarctic sea ice using Weddell Sea as a case study. To observe leads and their width with Sentinel-2 Level 1C products, it is necessary to have a surface-type classification. Therefore we analyzed Sentinel-2

**Table 3.** Different results from the literature and this study for the Weddell Sea sorted by publishing date. Threshold definition for lead identification differs between the studies. Marcq and Weiss (2012) use two different luminance thresholds. The last two entries are the results of this study for the Weddell Sea for which also two thresholds (OW: open water covered leads, OWN: open water and nilas covered leads) are applied. LF method stands for a linear fit and ML method stays for the method after Clauset et al. (2009). A detailed explanation of the methods is in Section 3.2.

| Source | Fitting method | Platform/ Instrument | Time and region | Resolution of the power law | Range of the power law | power-law exponent $\alpha$ |
|---|---|---|---|---|---|---|
| Wadhams (1981) | LF | submarine mission | October 1976, European Arctic Ocean | about 5 m | 50 - 1000 m | 2.00 |
| Wadhams et al. (1985) | LF | submarine mission | February 1967, Davis Strait | about 5 m | 50 - 1000 m | 2.29 |
| Lindsay and Rothrock (1995) | LF | AVHRR | 1989, Central Arctic Ocean | 1 km | 1 - 50 km | $1.60 \pm 0.18$ |
| Marcq and Weiss (2012) | ML | SPOT | April 1996, Central Arctic Ocean | 10 m | 0.02 - 2 km | 2.1 - 2.3 2.5 - 2.6 |
| Wernecke and Kaleschke (2015) | ML | CryoSat-2 | winter 2011 - 2014, Arctic Ocean | 300 m | $\geq 600$ m | $2.47 \pm 0.04$ |
| Qu et al. (2019) | LF | MODIS, Landsat-8 | April 2015, Beaufort Sea | 30 m - 1 km | $\geq 30$ m | 2.241 - 2.346 |
| this study | LF | Sentinel-2 | 2016 - 2018 (Nov - Apr), Weddell Sea | 10 m | 0.01 - 6.5 km | OW: $1.110 \pm 0.020$ OWN: $1.280 \pm 0.020$ |
| this study | ML | Sentinel-2 | 2016 - 2018 (Nov - Apr), Weddell Sea | 10 m | 0.01 - 6.5 km | OW: $1.399 \pm 0.002$ OWN: $1.413 \pm 0.002$ |

Level 1C products (band 4: 665 nm) with a resolution of 10 m and created a surface-type classification based on the top-of-the-atmosphere (TOA) reflectance. With this classification the Sentinel-2 Level 1C data can be used to detect and observe sea-ice leads under cloud-free conditions with a resolution of 10 m. The local overpass time of the two Sentinel-2 satellites matches the SPOT satellite and is close to Landsat 8, which provides the possibility for a future combination the data sets to longer time series. The mission lifetime for Sentinel-2 satellites, which were launched in 2015 and 2017, is planned to be 15 years (Drusch et al., 2012).

We apply two different fitting methods to the measured lead widths, which have been used in previous studies for Arctic sea ice (Wadhams, 1981; Wadhams et al., 1985; Lindsay and Rothrock, 1995; Marcq and Weiss, 2012; Wernecke and Kaleschke, 2015). The first fitting method is a linear fit (LF method), while the second method is based on a maximum likelihood approach by Clauset et al. (2009) (ML method). To further investigate influences on the power-law exponent, we define two different lead thresholds: OW for open water covered leads and OWN for open water and nilas covered leads. We confirm that the lead-width distribution for Weddell Sea sea ice follows a power law, showing similar behavior to the lead-width distribution in the Arctic, but with a smaller exponent. We also demonstrate that the fitting method has an influence on the result of the exponent and for further investigations, established methods should be applied to guarantee comparability of the results. With the LF method the power-law exponent for the lead-width distribution is 1.110 - 1.280 including both thresholds, while the exponent with the ML method shows less dependence on the threshold and is 1.399 - 1.413.

Thus, it is necessary to do further research on leads in the Southern Ocean to fully understand differences and similarities between the Arctic and Antarctic sea ice and account for possible regional differences in lead-widths throughout the Antarctic sea ice. For future comparison the same fitting method should be applied, since our study shows that with the same data different results occur.

*Data availability.* Analysed Sentinel-2 Level-1C products

All used Sentinel-2 Level-1C products are displayed in Table 1. We accessed the data using the Copernicus Open Access Hub (https://scihub.copernicus.eu/dhus/#/home).

*Author contributions.* MM acquired and checked the data, created the surface-type classification and derived the lead-width distribution under the supervision of LK. AS helped with the derivation of the lead width distribution and editing the paper. MM prepared the paper with contributions of all co-authors.

*Competing interests.* The authors declare that they have no conflict of interest.

*Acknowledgements.* This work was financially supported by the German Science Foundation (DFG) with the project number 314651818. The authors acknowledge the Copernicus program and the European space agency (ESA) for providing the imagery data for the Sentinel-2 satellites with the Copernicus Open Access Hub.

We thank the editors Yevgeny Aksenov and Jennifer Hutchings and the anonymous referees for their helpful criticism.

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
