# Peer review of "A lead-width distribution for Antarctic sea ice: a case study for the Weddell Sea with high resolution Sentinel-2 images"

_The Cryosphere, 2020_

## Referee Comment (RC1) · Anonymous Referee #1 · 15 Sep 2020

The manuscript aims to compare lead-width distribution estimates from an Arctic setting to an Antarctic setting. Within the manuscript a set of lead-width distribution algorithm parameter estimates are being calculated using a set of Sentinel-2 images. The algorithm parameter was found to be lower compared to the estimates for the Arctic sea ice areas. As the Antarctic sea ice areas are significantly less investigated compared to the Arctic sea ice areas the subject is interesting and timely. Unfortunately, the manuscript is poorly written and it's very unclear what the overall aim and the novel scientific contribution is.

Specific comments

[Figure]

It would be beneficial if the method is also used on a set of images covering the Arctic sea ice to see if the change in parameter setting relates to the Antarctic conditions, the method or the satellite data. In the Abstract it is stated that you compare exponents from the Arctic sea ice that do not agree with each other to your estimates for the Antarctic sea ice. Why do the estimates from the Arctic sea ice not agree?

As far as I can understand the nilas areas include both open water and nilas. Please confirm. It is unclear why these two lead types are not separated from each other, and if this separation was done in the scientific work by others that the lead parameter settings are compared to. This needs to be clarified.

Why are leads with dark-gray sea ice (up to 10cm thickness?) and light-gray (up to 30 cm thickness?) sea ice not included in the analysis? It is specifically stated in the methods section that you are investigating threshold for four different sea ice types. Do the studies compared to within this paper include those other sea ice types?

A schematic figure outlining the processing steps would aid the reader to understand the different steps within your method. How are the classified images validated? How many different individual leads were used in this study?

Overall the manuscript is poorly focused, many sentences difficult to follow and references to appropriate work is missing. The presented aim is the lead-width distribution however, the manuscript focuses on the algorithm parameter setting and not on the outcome of the lead-width estimates. Please either change the aim of the manuscript or change the manuscript to reflect the aim.

Technical corrections

Why was the Weddell sea picked for this study? Does it have a high/low frequency of leads compared to other areas of the Antarctic sea ice areas? Work by e.g. Willmes and Geinemann (2016) have indicated that different parts of the Arctic Ocean have different regional characteristics.

P1. R4-5. Unclear sentence please revise.

P1. R7-8 after the comma is a repeat of what was said on row 4-5. Please remove.

P1. R12. Replace bigger with larger or greater than

P1 R21. Please insert a reference.

P2 R34. How wide is a very narrow lead?

P3 R4-5. Unclear sentence please revise.

P3 R6. Insert references to these studies.

P3 R13-14. This is better suited in the discussion.

P3 R21. Are you using data from November to February or November to April? The way this is written now is confusing.

P4 R5. This section deals primarily with identifying thresholds and not classification of surface types. Consider changing the title. Moreover, the classification step is done in section 3.2 and should be reflected in the title there. I'm sorry but I could not follow how the lead width information was extracted from the images. Please clarify this.

P4 R9. Unclear sentence please revise.

P4 R16-18. Unclear sentence please revise.

P4 R28. Not only nilas but also open water.

P4. R30. What is apparent lead width? Please define.

Figure 2 Consider indicating the different TOA-reflectance areas that results in the different ice types, e.g. introduce a gray scale indicating classification areas. At the moment this figure is difficult to understand.

P5 Figure 3 text. What does "The swath does not cover the whole image area" mean? Either the Sentinel-2 image exist or not. Should image area be study area?

P4 R30 -P5 R1. Unclear sentence please revise.

P6. R2. What is x_width?

P6 R3. Where did the values for parameter C come from? This needs to be clarified.

P6 R13-14. Please add the 10m to the equation, or specify how the 10m were added to the equation.

P6 R17-18. This section is not results but should be put in the method or data section.

P6 R30. References missing.

P6 R29-30. Do the TOA values reported here also correspond to the values for the Arctic sea ice? The comparison here appears to be only for Antarctic data.

P7 R6. Compared to what?

Figure 5 is difficult to see. As the discussion primarily revolve around the for the two different methods for the two different mediums, why not combine the open water areas in one figure and indicate the different methods in the legend? Or better yet combine everything into one figure for easier interpretation. Consider also changing the color scheme as the colors are too similar and it makes it difficult to analyze the figure.

P8 R7-8. Unclear sentence please revise.

P9 R6-8. Unclear what you are trying to say here. Please discuss why different results are achieved and what this means.

P9 R24-26. According to Table 2 the resolution of the Wadhams papers are 5m. It is unclear what you are saying here. Please revise.

P9 R34-35. Please specify what these other data sources could be.

Section 4.3. This section does not belong in the results but should be moved to the data section instead.

Table 2. The table needs to be updated so that it is clear to the reader that the two values reported in the present study are for open water and nilas+open water areas. It is unclear from the present manuscript what the other reported values have used as criteria for their power law estimates? Are these values reflecting values for open water, nilas or leads generally? For the two different thresholds used in Marcq and Weiss (2012), do they also separate into two different lead types or are the two values a result of two different methods? If it is the latter please separate the values reported in the same way as is done with the present study results.

Reference Willmes S. and Heinemann G, (2016), Sea-Ice Wintertime Lead Frequencies and Regional Characteristics in the Arctic, 2003–2015, Remote Sensing, 8(1),4, https://doi.org/10.3390/rs8010004
* * *

---

## Referee Comment (RC2) · Anonymous Referee #2 · 11 Dec 2020

Comments on the manuscript "A lead-width distribution for Antarctic sea ice: a case study for the Weddell Sea with high resolution Sentinel-2 images", open for discussions in the Cryosphere Discussions.

The manuscript uses Sentinel-2 data from a visible channel (665 nm) to detect leads in the Weddell Sea during the daylight season between November 2016 and February 2018. The authors use an approach published in a previous study to identify the surface cover of the lead using thresholds of reflectance pertaining to each surface type, open water, nilas, grey ice, grey-white ice and first-year ice with snow cover. They also use another published approach to measure the apparent width of the lead as the

orientation can be different that the along-track direction of the satellite.

While the study provides interesting statistics about leads in the Weddell Sea, it does not offer details about the uncertainties implied in the data or the methodology (i.e. setting the thresholds). The subject is certainly timely and will have immediate implications, given the impacts of climate change on the polar regions, but I think with such a short manuscript it suits more a letter-style paper. I am not sure if TC allows this.

The manuscript is fairly well written though I found some sentence structure can be improved, e.g. "we noticed that on products with wide leads ...." or "The goal of the classification is to get thresholds . . .". A better word would be "to identify thresholds". The "amount of lead" should be "the number of leads", and so on. I am not including more details but the authors and the site editor(s) can adjust the style.

MAJOR ISSUES:

The title suggests that results of lead distribution apply to the Antarctic ice cover. I don't think this would true. The authors concentrate on the Weddell Sea as a case study. There are two features in this region that make it unique, the dominant old ice and the gyre (Weddell Sea gyre). That should make the leads in this area unique. Perhaps a better title should specify Weddell Sea only.

The authors admit that in presence of snow, the reflectance measurement will not an indicator of the underlying ice cover. I don't see how they addressed this problem. Please clarify.

While leads are mostly covered with open water or thin ice, they also serve as a path for broken thick ice. The authors set thresholds on reflectance to identify the surface of the lead, OW, nilas, grey ice, etc. If I understand it correctly, the authors calculate the lead statistics for each type separately (Fig. 5 shows OW and nilas), but a lead may have all types of ice plus OW. What if the lead in your data is composed of the five ice types you referred to? Would that be included in the lead statistics? The answer is still

not clear to me after reading the entire manuscript.

The authors use Reflectance to set thresholds to identify surface types. However, Reflectance is angular-dependent measurement, therefore it is not a property of the surface. Albedo is a property of the surface because it is the integration of the angular reflectance. The comparison between albedo and reflectance on page 6 lines 24-26 cannot be used to confirm the reflectance thresholds. To proceeds with the reflectance in this study, all leads must be viewed with the same angle from the different satellite orbits.

MINOR ISSUES:

INTRODUCTION: There are many more references that can be used to review the work on lead width geometry and width distributions. The authors should check and quote more than the 2 references used here. Moreover, the Introduction jumps too much between different themes. I would recommend the authors to re-write while grouping themes in separate paragraphs.

P 1 L18: leads may not always form in linear shape. P2 L1: what is "coupled climate models"? P2 L15: "Reiser et al. (2020) introduced a retrieval algorithm for lead fraction in the Antarctic, but these studies indicate that knowing about lead-width distributions is beneficial when estimating heat transfer". It is not clear how this can be a conclusion from a lead identification study. For one thing, it can only be a conclusion from a heat flux study but more importantly is a logical conclusion. It does not need a study. P2 L28: "To narrow down the effect of the fitting methods, we applied ...". Please change the sentence as the purpose of applying the two methods is not clear. P3 L4: "The determined thresholds for leads covered with open water .." Threshold on what? Reflectance? P3 L5: the sentence "Two threshold are applied to identify .." is not understood. Please rephrase. Figure 1: I find some of marked frames of the satellite images odd (with 3 or 5 sides instead of 4). Why is that? Also, in the caption, instead of "with shelf ice" it is better to use "including shelf ice border measured with . . ." then
specify the satellite radar sensor

DATA: P2 L10: attach the spatial resolution to the band. P2 L11: please include the spectral band for channel 4 (665 nm). Also, in Line 12, provide a reference to support your statement - that it is best for thin detection. I am not sure about the reason. Please specify. P2 L13: which Landsat? When I read this statement, I expect to see use of the coincident SPOT and LANDSAT data to support your finding. But they are mentioned here only to refer to their future use. Using these data will add value to the manuscript.

Table A1: why A1? It is not in an appendix? I would recommend inserting in the main text at the end of Section 2.

METHODS: The use of n=2 and n-3 in Equation 1 for light-grey ice and dark-grey ice is not clear to me. I see clearly the use of n=1 for the other types in Figure 2. I don't think there is something wrong here. It is just my failure to understand. Please re-phrase.

Also, according to my information, what you call dark-grey and light-grey ice, should be grey ice (10-15 cm thick) and grey-white ice (15-30 cm thick). These are nomenclature set by WMO and used in operational ice centers. You can refer to the document that defines ice types, published by the Canadian Ice service: MANICE (2005), "Manual of Standard Procedures for Observing and Reporting Ice Conditions", Canadian Ice Service – Environment Canada, ISBN 0-660-62858-9, Catalogue No. EN56-175/2005.

Figure 3: in the caption "The upper border of every image is 50 km wide". This is not the swath. Please clarify what you mean by upper border. This figure shows the OW in the lead. What about other ice cover, nilas, grey ice, etc.?

RESULTS P6 L21: the sentence "the TOA reflectance is only measured passively in the absence of clouds ..." is not clear. Please rephrase. P7 L5: Nilas threshold cover also open water? Why? There is overlap between nilas and OW, but the threshold does not cover both. P7 L6-21: it would be better to expand on the difference between lead statistics in the Arctic and Antarctic in a separate sub-section. Most of the lead studies

in the Arctic are conducted in the Beaufort Sea area. Some organized presentation should show the differences in the results, with related reasons if possible, and then comparison with findings from the Weddell Sea area.

CONCLUSION Before you mention about the method, you should mention the data used in the study.

---

## Editor Comment (EC1) · Jennifer Hutchings (Editor) · 6 Jan 2021

Dear Mr. Muchow, Dr. Schmitt and Lars,

I have taken over the editing process for your paper from Yvegenny Aksenov. Thank you for your contribution. Measuring lead widths and understanding their distribution is important for improving model representation of ocean-atmosphere fluxes. Studies like yours are valuable, and I hope you can see from the comments on the paper that your work might be a valuable contribution, providing high-resolution data where it has not been previously available. However there are some significant issues in the methodology and reporting that I hope you are able to address.

[Figure]

Two reviewers have provided a critique of the paper, and I invite you to respond to their reviews. There are specific points made that must be addressed. The paper could do with some rewriting to improve clarity. I agree that there are places the language can be improved for accuracy, so that your methodology and results are clear. I provide some examples of where significant language improvements are required, as do the reviewers. I would highly recommend asking an English speaking colleague to proof read the paper and point out where your language is either overly technical (and therefore obtuse to the layreader) or where you could clarify your point.

The first paragraph of the paper demonstrates some of the language that could be tightened and "Leads are created by dynamic motions of the sea ice (Miles and Barry, 1998) and thus follow a linear-like shape" Please rephrase, this is not correct.

"For parametrization of the lead width lead-width distributions are used." This could be clarified somewhat.

"Shear and divergence rates, estimated from models and satellite observations, follow a power law for the Arctic sea ice (e.g. Girard et al., 2009; Marsan et al., 2004), which suggests that a power law can also be used for the statistical parametrization of the lead width." Perhaps this paragraph would read more clearly if the second part was before this sentence. The mixing of observation types and models then makes the opening to the next paragraph confusing. There you want to clarify you are only talking about lead width observations, rather than "these studies".

Specify the wavelength of band 4.

"For the simplification of the problem only cloud-free Level-1C products were used". Can you even use your method for cloudy images. I think the statement "For the simplification" is not the correct phrase here.

"The goal of the classification is to get thresholds for different surface types". I think "get" is not the correct word to use here. What parameters are you identifying thresholds in? First state that the method is classification of surface types using x and y data. Then point out that you look for thresholds in band 4 reflectance that distingous different the surface types.

"Therefore, nine out of 20 later used Sentinel-2". Therefore is not the correct word to use here. There is no follow on from the previous sentence.

I agree with the reviewers that the methods section would benefit from some additional figures to demonstrate the procedure and how thresholds are identified. Some particular comments: Where do the values in Table 1 come from? Is your Gaussian fit appropriate? Are the thresholds consistent across images with different sun and look angles? I also agree that you can clarify the methods description. For example you could describe what the apparent width is succintly, and how this differs from the true width.

Figures 4 and 5 lead me to wonder if the apparent lead width can be accurately measured down to 10m. There is an arc is the data points, with the gradient becoming less steep towards smaller lead widths, which could be indicative of white noise. I echo the reviewers interest in a more detailed discussion of the errors introduced in the method.

Can you describe the reasons for the differences in LF and MF fit based on their weighting of the data? It might be more interesting to consider if the fit is statistically significant.

"Based on our current knowledge, previous studies (Table 2) focused on different regions in the Arctic and not on Antarctic regions." Here "Based on our current knowledge" is not needed in the sentence.

When comparing to other studies in the Arctic it should be clarified that you are not comparing like method for like. In particular the different sensors have different spatial resolution, which will constrain the range over which the power law fit is made. If you consider how there appears to be a decrease in exponent at smaller scales in your

data, would this explain any of the differences to previous studies with lower resolution sensors?

I think your study highlights the need for more studies like yours, providing more data on lead width. It is confusing that there is so much spread in power law exponents between these studies, and whether this is related to differences in lead detection method and the physical processes controlling lead width is hard to determine.

Finally, it is my understanding that The Cryosphere accepts short contributions. However, unlike one reviewer, I believe your paper could be strengthen by including a little more content in the methodology, to ensure clarity of your approach and reproducibility. Including a consideration of errors and the robustness of your results and whether the fits are significant would greatly improve the paper. You could also provide more guidance in the discussion to address the reviewers concerns regarding the scientific contribution you make.

At this stage I believe the paper requires major revisions, however the revisions might change the main findings which would require a second review of the paper and further consideration. Please let me know if you need any clarity on this. And I look forward to your response to the reviews.

With very best regards. Jenny

---

## Author Comment (AC1) · 27 Feb 2021

**Answer to Anonymous Referee 1**

*We thank the editor Dr. Jenny Hutchings and the two anonymous referees for the helpful comments and their efforts reviewing the paper. Please find point-by-point answers to the comments below. The answers to the other referee and the editors note are separate comments.*

**Anonymous Referee 1**

The manuscript aims to compare lead-width distribution estimates from an Arctic set-

ting to an Antarctic setting. Within the manuscript a set of lead-width distribution algorithm parameter estimates are being calculated using a set of Sentinel-2 images. The algorithm parameter was found to be lower compared to the estimates for the Arctic sea ice areas. As the Antarctic sea ice areas are significantly less investigated compared to the Arctic sea ice areas the subject is interesting and timely. Unfortunately, the manuscript is poorly written and it's very unclear what the overall aim and the novel scientific contribution is.

Specific comments

It would be beneficial if the method is also used on a set of images covering the Arctic sea ice to see if the change in parameter setting relates to the Antarctic conditions, the method or the satellite data. In the Abstract it is stated that you compare exponents from the Arctic sea ice that do not agree with each other to your estimates for the Antarctic sea ice. Why do the estimates from the Arctic sea ice not agree?
*The estimates from the Arctic sea ice do not agree to each other as discussed in the manuscript. It is unclear if this is related to the fitting method or the underlying data set. For future comparisons the same fitting method should be applied to estimate the exponents. We agree that it would be interesting to analyze images from the Arctic. However, such an analysis was not within the scope of this study. We will include this point in the discussion an elaborate more on the (regional) differences of the exponents.*

As far as I can understand the nilas areas include both open water and nilas. Please confirm. It is unclear why these two lead types are not separated from each other, and if this separation was done in the scientific work by others that the lead parameter settings are compared to. This needs to be clarified.
*Yes, nilas is both open water and nilas. We do not separate these from each other since the water in leads refreezes quickly depending on the surrounding temperatures. But even when leads are covered with thin ice like nilas, they have properties that are characteristic for leads with regard to heat exchange, ice production and also being*

*navigable by several surface vessels (see "WMO Sea Ice Nomenclature (WMO No.259, volume 1", 7.3 Lead). We will make that clearer in the new version by renaming the two thresholds and explaining the use of both thresholds. https:// library.wmo.int/ index. php?lvl=notice_display&id=6772#.YAgyIsJ7mUk (last access: 20.01.2021)*

Why are leads with dark-gray sea ice (up to 10cm thickness?) and light-gray (up to 30 cm thickness?) sea ice not included in the analysis? It is specifically stated in the methods section that you are investigating threshold for four different sea ice types. Dot he studies compared to within this paper include those other sea ice types?
*The surface classification are not connected to any measured thicknesses, since the Sentinel-2 Level-1C product only provides optical data, which is why the names correspond to the "grayness" of the ice. For the calculation of the lead with distribution we decided to focus on leads covered by open water or nilas only – mainly because previous studies also investigated leads with these surface types. This is necessary to make the obtained results more comparable. The other surface types could be used for different applications. We will clarify this in the corresponding sections.*

A schematic figure outlining the processing steps would aid the reader to understand the different steps within your method. How are the classified images validated? How many different individual leads were used in this study?
*In total, the number of leads is 2024 for the open water threshold and 3799 for the open water and nilas threshold. Thank you for the recommendation. We will create a schematic figure for describing the selection process of the images. We will include this one or a similar one in the data section (Section 2.) and will adapt the description of the method to enhance clarity. We also prepared two additional figures for the method section.*

Overall the manuscript is poorly focused, many sentences difficult to follow and references to appropriate work is missing. The presented aim is the lead-width distribution however, the manuscript focuses on the algorithm parameter setting and not on the outcome of the lead-width estimates. Please either change the aim of the manuscript

or change the manuscript to reflect the aim.

*We will revise the manuscript according to your detailed suggestions below. Regarding the focus, we will clarify that the ice type and threshold detection is only a prerequisite, which is necessary to identify leads. The main aim is then to use these leads to obtain a lead width distribution. We will adapt the corresponding section accordingly.*

Technical corrections

Why was the Weddell sea picked for this study? Does it have a high/low frequency of leads compared to other areas of the Antarctic sea ice areas? Work by e.g. Willmes and Heinemann (2016) have indicated that different parts of the Arctic Ocean have different regional characteristics.

*The Weddell Sea was chosen for its special scientific importance, its size, the size of the sea-ice cover during November to April and availability of cloud free images in the Southern Ocean. Furthermore, Sentinel-2 is a land mission and images over ocean are only available in the vicinity of land which restrict the regional selection. We will discuss that results from the Weddell sea are not necessarily representative for other Antarctic areas. We will include the Willmes and Heinemann (2016) paper for comparisons and further assessment of our results.*

P1. R4-5. Unclear sentence please revise. *Will be changed.*

P1. R7-8 after the comma is a repeat of what was said on row 4-5. Please remove. *Will be changed.*

P1. R12. Replace bigger with larger or greater than *Will be changed.*

P1 R21. Please insert a reference. *Will be fixed.*

P2 R34. How wide is a very narrow lead? *The "narrow lead" refers to the high resolution of 10 m of our study.*

P3 R4-5. Unclear sentence please revise. *Will be changed.*

P3 R6. Insert references to these studies. *Will be fixed.*

P3 R13-14. This is better suited in the discussion. *We agree.*

P3 R21. Are you using data from November to February or November to April? The way this is written now is confusing.
*The sentence will be rewritten. The data is always part of the months between November to April, while being from the overall time period of November of 2016 to February of 2018.*

P4 R5. This section deals primarily with identifying thresholds and not classification of surface types. Consider changing the title. Moreover, the classification step is done in section 3.2 and should be reflected in the title there. I'm sorry but I could not follow how the lead width information was extracted from the images. Please clarify this.
*The section 3.1. will be renamed in a way that indicates the topic of "threshold identification". The lead width information was conducted in a following way: 1.) A binary image was created with the threshold (see Figure 3), then a measurement grid was put on top of the image consisting of 10 vertical and horizontal measurement tracks, which have a distance of 10 km to each other. The lead width is then measured across each track with counting the pixels (black in Figure 3 for indicating a lead), which then refers to the width of the lead.*

P4 R9. Unclear sentence please revise. *Will be revised.*

P4 R16-18. Unclear sentence please revise. *Will be revised.*

P4 R28. Not only nilas but also open water. *Yes, the thresholds will be renamed for clarity.*

P4. R30. What is apparent lead width? Please define.
*The "true lead width" would be measured if every lead width was measured orthogonal to the lead. Our measurement grid can not guarantee that, which means that we sometimes measure not the shortest width but the width of a line across the lead with*

*an angle other than 90 degree. That is why we call the lead width "apparent lead width" similar to Wernecke and Kaleschke (2015) . This will be stated more clearly in the text.*

Figure 2 Consider indicating the different TOA-reflectance areas that results in the different ice types, e.g. introduce a gray scale indicating classification areas. At the moment this figure is difficult to understand.
*In this figure we do not show the classification of different ice types, but rather use a single threshold (open water or open water/nilas) to distinguish lead and sea ice areas. This results in a binary image (black for lead and white for ice) and thus introducing a gray scale colormap would not be helpful here. We will improve the figure caption to make this more clear.*

P5 Figure 3 text. What does "The swath does not cover the whole image area" mean? Either the Sentinel-2 image exist or not. Should image area be study area?
*The Sentinel-2 Level-1C products are always 100 km x 100 km products, but the swath of the satellite represented in the product does not always cover the whole area of the product, which is why the bottom right corner in the upper Band 4 image has a solid white triangle. To clarify, we will replace "image area" by "figure area".*

P4 R30 -P5 R1. Unclear sentence please revise. *Will be revised.*

P6. R2. What is $x_{width}$? *x_width is the measured apparent lead width.*

P6 R3. Where did the values for parameter C come from? This needs to be clarified.
*The parameter C is the offset at the y-axis of the function and is therefore related to the number of measurements.*

P6 R13-14. Please add the 10m to the equation, or specify how the 10m were added to the equation.
*The 10 m were added for the variable "step size", since the 10 m indicate the step size of the measured lead widths.*

P6 R17-18. This section is not results but should be put in the method or data section.

*In Section 4.1 we use the thresholds determined using the method described in 3.1 and compare the resulting reflectances to albedo values from the literature. Thus, this is rather a discussion part - which we decided to combine with the Results section to improve the readability of the manuscript.*

P6 R30. References missing. *Will be fixed.*

P6 R29-30. Do the TOA values reported here also correspond to the values for the Arctic sea ice? The comparison here appears to be only for Antarctic data.
*As we stated above, this would be interesting to investigate in the future but is beyond the scope of this study.*

P7 R6. Compared to what? Figure 5 is difficult to see. As the discussion primarily revolve around the for the two different methods for the two different mediums, why not combine the open water areas in one figure and indicate the different methods in the legend? Or better yet combine everything into one figure for easier interpretation. Consider also changing the color scheme as the colors are too similar and it makes it difficult to analyze the figure.
*We disagree with putting everything (4 lines, two data sets) into one plot, since the lines then tend to overlap and the figure is crowded and it becomes difficult to understand the fine differences. We decided to create 2 figure panels for each of the two question of the problem. 1) The influence of the applied fitting method on the same data set, which is Figure 4. Figure 4 shows the combination of both methods only for the open water threshold (left figure) and only for the open water and nilas threshold (right figure). 2) The influence of the used threshold on slightly different data sets, where we can see that for example the influence of the data set for the ML method is really small. These nuances would be more difficult to highlight in one combined image.*

P8 R7-8. Unclear sentence please revise. *Will be revised.*

P9 R6-8. Unclear what you are trying to say here. Please discuss why different results are achieved and what this means.

*Here, "results" refers to the different power law exponents in the literature. We will extend the discussion of the differences of the results and what this can mean for future investigations.*

P9 R24-26. According to Table 2 the resolution of the Wadhams papers are 5m. It is unclear what you are saying here. Please revise.
*The Wadhams studies did not use satellites, but submarine missions, thus their possible spatial coverage is smaller. The advantages of satellite data are the increased spatial coverage and repeated overpasses. Here, we wanted to highlight the high spatial resolution of Sentinel-2 and the possible advantages compared to other satellite products.*

P9 R34-35. Please specify what these other data sources could be.
*Since Sentinel-2 only provides optical information a combination with field measurements for the validation of the surface types would be beneficial.*

Section 4.3. This section does not belong in the results but should be moved to the data section instead.
*We will move this section to section 2. Methods.*

Table 2. The table needs to be updated so that it is clear to the reader that the two values reported in the present study are for open water and nilas+open water areas. It is unclear from the present manuscript what the other reported values have used as criteria for their power law estimates? Are these values reflecting values for open water, nilas or leads generally? For the two different thresholds used in Marcq and Weiss (2012), do they also separate into two different lead types or are the two values a result of two different methods? If it is the latter please separate the values reported in the same way as is done with the present study results.
*We will update the table to make the differences between the results more obvious. Marcq and Weiss (2012) used two different thresholds for leads (with open water) based on two different luminescence thresholds.*

Reference Willmes S. and Heinemann G, (2016), Sea-Ice Wintertime Lead Frequencies and Regional Characteristics in the Arctic, 2003–2015, Remote Sensing, 8(1),4,https://doi.org/10.3390/rs801

---

## Author Comment (AC2) · 27 Feb 2021

**Answer to Anonymous Referee 2**

*We thank the editor Dr. Jenny Hutchings and the two anonymous referees for the helpful comments and their efforts reviewing the paper. Please find point-by-point answers to the comments below. The answers to the other referee and the editors note are separate comments.*

**Anonymous Referee 2**

While the study provides interesting statistics about leads in the Weddell Sea, it does

not offer details about the uncertainties implied in the data or the methodology (i.e. setting the thresholds). The subject is certainly timely and will have immediate implications, given the impacts of climate change on the polar regions, but I think with such as short manuscript it suits more a letter-style paper. I am not sure if TC allows this.

*By including the suggestions by the referees, the paper will be more detailed and longer than before. Thus, we want to continue to handle the manuscript as a full paper and not change it into a letter.*

The manuscript is fairly well written though I found some sentence structure can be improved, e.g. "we noticed that on products with wide leads ...." or "The goal of the classification is to get thresholds...". A better word would be "to identify thresholds". The "amount of lead" should be "the number of leads", and so on. I am not including more details but the authors and the site editor(s) can adjust the style.

*Thank you, we will check the overall style of the manuscript.*

MAJOR ISSUES:

The title suggests that results of lead distribution apply to the Antarctic ice cover. I don't think this would true. The authors concentrate on the Weddell Sea as a case study. There are two features in this region that make it unique, the dominant old ice and the gyre (Weddell Sea gyre). That should make the leads in this area unique. Perhaps a better title should specify Weddell Sea only.

*The title should imply that this is the first lead-width distribution for an area (the Weddell Sea) of the Antarctic sea ice, therefore the label case study. Since both reviews highlighted a lack of explanation of the reasoning for the study area, we will specify that topic more through the paper.*

The authors admit that in presence of snow, the reflectance measurement will not an indicator of the underlying ice cover. I don't see how they addressed this problem. Please clarify.

*We assume that nilas has no or only a really thin snow cover, since snow would*
*brighten the appearance of the surface. Additionally, snow has an insulating effect and therefore inhibits the heat exchange for an example which would reduce the "lead effect" of an area where nilas is covered with snow. This is also the reason, why the names of our surface categories do not indicate any age of the ice or thickness of the ice, since – as you stated – the ice in the category "sea ice covered with snow" can be really different. But since we use the thresholds from the classification only for measuring lead widths, we are confident in the thresholds.*

While leads are mostly covered with open water or thin ice, they also serve as a path for broken thick ice. The authors set thresholds on reflectance to identify the surface of the lead, OW, nilas, grey ice, etc. If I understand it correctly, the authors calculate the lead statistics for each type separately (Fig. 5 shows OW and nilas), but a lead may have all types of ice plus OW. What if the lead in your data is composed of the five ice types you referred to? Would that be included in the lead statistics? The answer is still not clear to me after reading the entire manuscript.
*We only investigate leads with open water or leads with nilas and open water. The other surface categories are not used for the lead-width estimation. Also, the nilas threshold contains areas covered with nilas and open water. This then covers for example also the case of a lead, where one area starts to refreeze while the other part is still open. Thus, in that case we measure the whole lead width. We decided to omit leads that are covered by brighter (thicker) ice, since this has been common practice in many other studies (e.g. Marcq and Weiss (2016) and Lüpkes et al. (2008)) and thus makes our results more comparable.*

*Lüpkes, Christof, et al. "Influence of leads in sea ice on the temperature of the atmospheric boundary layer during polar night." Geophysical Research Letters 35.3 (2008).*

The authors use Reflectance to set thresholds to identify surface types. However, Reflectance is angular-dependent measurement, therefore it is not a property of the surface. Albedo is a property of the surface because it is the integration of the angular reflectance. The comparison between albedo and reflectance on page 6 lines 24-26

cannot be used to confirm the reflectance thresholds. To proceeds with the reflectance in this study, all leads must be viewed with the same angle from the different satellite orbits.

*This is an appropriate simplification because it is not practical to restrict the observed reflection angle. The reflectance anisotropy could in principle be estimated using many observations over rather static scenes. However, we do not expect a significant influence of a potential anisotropy on the resulting lead with distribution. We estimated the thresholds with images from January to April of 2017. Thus, the data for fitting the Gaussian curves includes several sun and look angles. Before estimating the thresholds we compared the TOA reflectance values for the the surface types within the each products and found no significant difference, but nevertheless used nine products for the classification for a larger data base.*

MINOR ISSUES:

INTRODUCTION: There are many more references that can be used to review the work on lead width geometry and width distributions. The authors should check and quote more than the 2 references used here. Moreover, the Introduction jumps too much between different themes. I would recommend the authors to re-write while grouping themes in separate paragraphs.
*Thank you for the suggestion, we will rewrite the Introduction accordingly.*

P 1 L18: leads may not always form in linear shape.
*We will replace "thus" by "often" to also include other possible shapes.*

P2 L1: what is "coupled climate models"?
*"coupled climate models" commonly refers to models consisting at least of an ocean or land and an atmosphere component which are coupled to each other.*

P2 L15: "Reiser et al. (2020) introduced a retrieval algorithm for lead fraction in the Antarctic, but these studies indicate that knowing about lead-width distributions is beneficial when estimating heat transfer". It is not clear how this can be a conclusion from

a lead identification study. For one thing, it can only be a conclusion from a heat flux study but more importantly is a logical conclusion. It does not need a study.

*The mention of Reiser et al. (2020) was supposed to highlight that there are lead-fraction products available, also for the Southern Ocean, but that the lead width is still missing. And that other studies (the ones named before) have indicated, that the lead width is important. We will clarify that part.*

P2 L28: "To narrow down the effect of the fitting methods, we applied ...". Please change the sentence as the purpose of applying the two methods is not clear. *Will be revised.*

P3 L4:"The determined thresholds for leads covered with open water .." Threshold on what?Reflectance? *Yes, the threshold is a TOA reflectance threshold.*

P3 L5: the sentence "Two threshold are applied to identify .." is not understood. Please rephrase. *Will be revised.*

Figure 1: I find some of marked frames of the satellite images odd (with 3 or 5 sides instead of 4). Why is that? Also, in the caption, instead of "with shelf ice" it is better to use "including shelf ice border measured with..." then specify the satellite radar sensor *The form of the products is due to the differences of the satellite swath and the processing grid of the ESA which results in areas without a satellite measurement in the image (as explained and shown in Figure 3 with the missing corner in the bottom right corner in the upper image). We decided to use the "real image outlines" for the map to show the product size.*

DATA:

P3 L10: attach the spatial resolution to the band.
*The resolution of each band is not directly indicated, because it is irregularly distributed, and we only use the 10 m resolution of Band 4. See: https:// sentinel.esa.int/ web/ sentinel/ user-guides/ sentinel-2-msi/ resolutions/ spatial*

P3 L11: please include the spectral band for channel 4 (665 nm). *Will be changed.*

Also, in Line 12, provide a reference to support your statement - that it is best for thin detection. I am not sure about the reason. Please specify.
*We manually compared Band 2 – 4 and 8 with each other in areas, where we expected thin ice in leads. After visual inspection we and decided to use Band 4. We will specify this in the text.*

P3 L13: which Landsat? When I read this statement, I expect to see use of the coincident SPOT and LANDSAT data to support your finding. But they are mentioned here only to refer to their future use. Using these data will add value to the manuscript.
*This information is a citation from Drusch et al. (2012), but at least Landsat 8 crosses the Equator every day at 10 a.m., while Sentinel-2 crosses the Equator at 10:30 am. The scope of the study was only to use Sentinel-2, this statement is just an outlook and will be moved to the end of the manuscript – according to the suggestion by reviewer 1*

Table A1: why A1? It is not in an appendix? I would recommend inserting in the main text at the end of Section 2.
*We decided to put the table in the appendix due to its size, but we are also fine with placing it directly into Section 2.*

METHODS:

The use of n=2 and n-3 in Equation 1 for light-grey ice and dark-grey ice is not clear to me. I see clearly the use of n=1 for the other types in Figure 2. I don't think there is something wrong here. It is just my failure to understand. Please re-phrase.
*The n indicates the number of Gaussian curves which were combined to one function for fitting the histograms. By using n>1 we can account for multiple maxima within a distribution. We will re-phrase that sentence.*

Also, according to my information, what you call dark-grey and light-grey ice, should be grey ice (10-15 cm thick) and grey-white ice (15-30 cm thick). These are nomenclature

[Figure]

set by WMO and used in operational ice centers. You can refer to the document that defines ice types, published by the Canadian Ice service: MANICE (2005), "Manual of Standard Procedures for Observing and Reporting Ice Conditions", Canadian Ice Service – Environment Canada, ISBN 0-660-62858-9, Catalogue No. EN56-175/2005. *Since we only have information about the brightness of the ice and not of the thickness, we created our own terminology to indicate the brightness of the ice. However, we like your suggestion of using the WMO standard nomenclature. We will add a short explanation indicating that our terminology does not refer to ice thickness but rather optical properties.*

Figure 3: in the caption "The upper border of every image is 50 km wide". This is not the swath. Please clarify what you mean by upper border.
*We mean that the width of the figure corresponds to 50 km, thus we will change the text to "The area shown in the figure has a width of 50 km."*

This figure shows the OW in the lead. What about other ice cover, nilas, grey ice, etc.?
*Figure 3 is used to illustrate the lead detection algorithm desribed in Section 3.2. The figure on the right shows the leads for the nilas and open water threshold. With that, the binary image for both applied thresholds are shown. The other surface types are not used for lead-width measurements, which is why we did not show them initially. We agree that it would be beneficial to also show an example of the classification described in Section 3.1 showing all 5 surface types.*

RESULTS

P6 L21: the sentence "the TOA reflectance is only measured passively in the absence of clouds ..." is not clear. Please rephrase. *Will be changed.*

P7 L5: Nilas threshold cover also open water? Why? There is overlap between nilas and OW, but the threshold does not cover both.
*The nilas threshold covers both. We realized that this is not clear in the manuscript. We will change that with renaming the thresholds in a more clear way.*

P7 L6-21: it would be better to expand on the difference between lead statistics in the Arctic and Antarctic in a separate sub-section. Most of the lead studies in the Arctic are conducted in the Beaufort Sea area. Some organized presentation should show the differences in the results, with related reasons if possible, and then comparison with findings from the Weddell Sea area.
*We will restructure the discussion and give additional detailed about the differences.*

CONCLUSION

Before you mention about the method, you should mention the data used in the study.
*Will be revised.*

---

## Author Comment (AC3) · 27 Feb 2021

*Dear Dr. Jenny Hutchings,*

*thank you for handling the editing process and for giving all this hands-on advice for improvement of the paper. Our responses to the referees are posted. If anything is missing and/or unclear, we are happy to provide more answers and information. We are thankful for the explicit examples for language improvement made by you and the reviewers and it was helpful to see where we can improve our writing. We also agree, that the paper will become longer after we edit it due to integration of more content in the method section and discussion of our results and that a short letter-styled paper*

[Figure]

*will not benefit this paper.*

*Thank you for your time and kind regards, Marek Muchow, Amelie Schmitt Lars Kaleschke*

**Specific Editor comments**
The first paragraph of the paper demonstrates some of the language that could be tightened and "Leads are created by dynamic motions of the sea ice (Miles and Barry,1998) and thus follow a linear-like shape" Please rephrase, this is not correct. *Will be revised.*

"For parametrization of the lead width lead-width distributions are used." This could be clarified somewhat. *We will revise this sentence for clarity.*

"Shear and divergence rates, estimated from models and satellite observations, follow a power law for the Arctic sea ice (e.g. Girard et al., 2009; Marsan et al., 2004), which suggests that a power law can also be used for the statistical parametrization of the lead width." Perhaps this paragraph would read more clearly if the second part was before this sentence. *Will be revised.*

The mixing of observation types and models then makes the opening to the next paragraph confusing. There you want to clarify you are only talking about lead width observations, rather than "these studies". *Will be revised.*

Specify the wavelength of band 4. *The wavelength is 665 nm.*

"For the simplification of the problem only cloud-free Level-1C products were used". Can you even use your method for cloudy images. I think the statement "For the simplification" is not the correct phrase here. *Indeed, the method only works for cloud-free cases. We will rephrase this sentence for clarity.*

"The goal of the classification is to get thresholds for different surface types". I think "get" is not the correct word to use here. *Will be revised.*

What parameters are you identifying thresholds in? First state that the method is classification of surface types using x and y-data. Then point out that you look for thresholds in band 4 reflectance that distinguish different the surface types. *We will rephrase the description of the method for clarity.*

"Therefore, nine out of 20 later used Sentinel-2". Therefore is not the correct word to use here. There is no follow on from the previous sentence. *We will rephrase this sentence for clarity.*

I agree with the reviewers that the methods section would benefit from some additional figures to demonstrate the procedure and how thresholds are identified. *We will add schematics to clarify the methodology.*

Some particular comments:

Where do the values in Table 1 come from?
*The values are derived from the intersection points of the Gaussian curves for different surface types, as described in Section 3.1. We will include a more detailed reference to Table 1 in the text.*

Is your Gaussian fit appropriate?
*We fitted the Gaussian curves to histogram data, which indicated a Gaussian distribution or a combination of Gaussian distributions with several maxima.*

Are the thresholds consistent across images with different sun and look angles?
*We estimated the thresholds with images from January to April of 2017. Thus, the data for fitting the Gaussian curves includes several sun and look angles. Before estimating the thresholds we compared the TOA reflectance values for the the surface types within the each products and found no significant difference, but nevertheless used 9 products for the classification for a larger data base.*

I also agree that you can clarify the methods description. For example you could describe what the apparent width is succinctly, and how this differs from the true width. *As suggested by the reviewers, we will add a more detailed description.*

Figures 4 and 5 lead me to wonder if the apparent lead width can be accurately measured down to 10m. There is an arc is the data points, with the gradient becoming less steep towards smaller lead widths, which could be indicative of white noise.

*We have actually measured leads with a with of 10 m, but as stated, the amount of leads with a width of 10 m is less then might be expected (see Figures 4 and 5). A possible reason is the resolution itself, since capturing a 10 m lead is more difficult than a 30 m lead with a resolution of 10 m. To accurately capture a 10 m lead the lead needs to be directly in the same place as the "image pixel". This problem is also explained by Wernecke and Kaleschke (2015) "Compared to the power law, the found number of apparent lead width of 300 m is smaller than expected. This is a typical feature of the lower bound of the resolution as leads of this size are not always covered by a single measurement but partially by more, not necessarily leading to a detection". We will add a short discussion in the paper.*

I echo the reviewers interest in a more detailed discussion of the errors introduced in the method.
*We will include an extended error discussion were a lack of discussion was identified through the review process.*

Can you describe the reasons for the differences in LF and MF fit based on their weighting of the data?
*The linear fit (LF) method applies a least-square approach, which means that every data point has the same distance to the best fit. The MF method by Clauset et al. (2009) is based on a maximum likelihood approach.*

It might be more interesting to consider if the fit is statistically significant. For both fits, we calculated the standard error of the slope/fitted parameter, which we called standard deviation in the paper. While revisiting, we realized that the term "standard deviation" alone might be not explanatory enough.
*We will discuss the significance in more detail.*

"Based on our current knowledge, previous studies (Table 2) focused on different regions in the Arctic and not on Antarctic regions." Here "Based on our current knowledge" is not needed in the sentence. *Will be revised.*

When comparing to other studies in the Arctic it should be clarified that you are not comparing like method for like. In particular the different sensors have different spatial resolution, which will constrain the range over which the power law fit is made.
*Yes, as suggested by the reviewers, we will clarify the differences of the sensors and methods used in the literature.*

If you consider how there appears to be a decrease in exponent at smaller scales in your data, would this explain any of the differences to previous studies with lower resolution sensors?
*We will add a discussion about a possible scale dependency of the power law fit.*

---

## Author Response (AR1)

Dear Yevgeny Aksenov, Jenny Hutchings and Referees,

thank you for giving all the hands-on advice for improving the paper. All changes are highlighted in bold text within the track-changes file. Here, we want to present an overview of the major changes for the second manuscript:

- We restructured the introduction, discussion and conclusions of the paper
- We included three different flow charts to explain the data selection and processing (Figure 1, Figure 2 and Figure 3) in Section 2 and 3. Therefore, we made changes within Figure 1, added Figure 2 and replaced Figure 3 with a completely new figure Section 3.1 was then rewritten and restructured to fit the flow charts
- We changed the names of the sea-ice types used for the surface-type classification according to the WMO Sea Ice Nomenclature
- We gave new names to the two used TOA-thresholds
- We re-calculated the power-law exponents and their standard deviation (see Section 3.2.)
- We extended the discussion in regards to the difference between the values for the Arctic and in regards to our results for the Weddell Sea and their possible representation for the Antarctic sea ice
- We removed Section 4.3 and shifted it contents to Data (Section 2) and the discussion (Section 4)

A point-by-point response to the Editor Comment and the Referees can be found in our answers (AC1 – AC3), where we also indicated our planned and then realized changes.

Thank you for your time and kind regards,
Marek Muchow, Amelie Schmitt, Lars Kaleschke

---

## Author Response (AR2)

*We thank the editors Jenny Hutchings and Yevgeny Aksenov and the two anonymous referees for the helpful comments and their efforts reviewing the paper.*
*The major changes within the second manuscript are the following:*

*• We restructured the introduction, discussion and conclusions of the paper*
*• We included three different flow charts to explain the data selection and processing (Figure 1, Figure 2 and Figure 3) in Section 2 and 3. Therefore, we made changes within Figure 1, added Figure 2 and replaced Figure 3 with a completely new figure Section 3.1 was then rewritten and restructured to fit the flow charts*
*• We changed the names of the sea-ice types used for the surface-type classification according to the WMO Sea Ice Nomenclature*
*• We gave new names to the two used TOA-thresholds (open water (ON) and open water and nilas (OWN) threshold)*
*• We re-calculated the power-law exponents and their standard deviation (see Section 3.2.)*
*• We extended the discussion in regards to the difference between the values for the Arctic and in regards to our results for the Weddell Sea and their possible representation for the Antarctic sea ice*
*• We removed Section 4.3 and shifted it contents to Data (Section 2) and the discussion (Section 4)*

*Please find point-by-point answers to the comments below. The answers to the other referee and the editors note are separate comments.*

**Specific Editor comments**

The first paragraph of the paper demonstrates some of the language that could be tightened and "Leads are created by dynamic motions of the sea ice (Miles and Barry,1998) and thus follow a linear-like shape" Please rephrase, this is not correct. *Revised.*

"For parametrization of the lead width lead-width distributions are used." This could be clarified somewhat. *Revised.*

"Shear and divergence rates, estimated from models and satellite observations, follow a power law for the Arctic sea ice (e.g. Girard et al., 2009; Marsan et al., 2004), which suggests that a power law can also be used for the statistical parametrization of the lead width." Perhaps this paragraph would read more clearly if the second part was before this sentence. *Revised.*

The mixing of observation types and models then makes the opening to the next paragraph confusing. There you want to clarify you are only talking about lead width observations, rather than "these studies". *We reorganized some parts of the introduction.*

Specify the wavelength of band 4. *Edited (665 nm)*

"For the simplification of the problem only cloud-free Level-1C products were used". Can you even use your method for cloudy images. I think the statement "For the simplification" is not the correct phrase here. *Indeed, the method only works for cloud-free cases. We removed "for the simplification… ".*

"The goal of the classification is to get thresholds for different surface types". I think "get" is not the correct word to use here. *Changed.*

What parameters are you identifying thresholds in? First state that the method is classification of surface types using x and y-data. Then point out that you look for thresholds in band 4 reflectance that distinguish different the surface types. *We rephrased and restructured the method section. Additionally, we created schematic figures to outline the process.*

"Therefore, nine out of 20 later used Sentinel-2". Therefore is not the correct word to use here. There is no follow on from the previous sentence. *We rephrased the sentence and restructured the text parts around it.*

I agree with the reviewers that the methods section would benefit from some additional figures to demonstrate the procedure and how thresholds are identified. *We added schematics to clarify the methodology.*

Some particular comments:

Where do the values in Table 1 come from?
*The values are derived from the intersection points of the Gaussian curves for different surface types, as described in Section 3.1. We will include a more detailed reference to Table 1 in the text.*

Is your Gaussian fit appropriate?
*We fitted the Gaussian curves to histogram data, which indicated a Gaussian distribution or a combination of Gaussian distributions with several maxima.*

Are the thresholds consistent across images with different sun and look angles?
*We estimated the thresholds with images from January to April of 2017. Thus, the data for fitting the Gaussian curves includes several sun and look angles. Before estimating the thresholds we compared the TOA reflectance values for the the surface types within the each products and found no significant difference, but nevertheless used 9 products for the classification for a larger data base.*

I also agree that you can clarify the methods description. For example you could describe what the apparent width is succinctly, and how this differs from the true width.
*As suggested by the reviewers, we added a more detailed description.*

Figures 4 and 5 lead me to wonder if the apparent lead width can be accurately measured down to 10m. There is an arc is the data points, with the gradient becoming less steep towards smaller lead widths, which could be indicative of white noise.
*We have actually measured leads with a with of 10 m, but as stated, the amount of leads with a width of 10 m is less then maybe expected (see Figures 4 and 5). A reason therefore is the resolution itself, since capturing a 10 m lead is more difficult than a 30 m lead with a resolution of 10 m. To accurately capture a 10 m lead the lead needs to be directly in the same place as the "image pixel". This problem is also explained by Wernecke and Kaleschke (2015) "Compared to the power law, the found number of apparent lead width of 300 m is smaller than expected. This is a typical feature of the lower bound of the resolution as leads of this size are not always covered by a single measurement but partially by more, not necessarily leading to a detection". We included this discussion in the paper.*

I echo the reviewers interest in a more detailed discussion of the errors introduced in the method.
*We recalculated the lead-width distribution power law exponents and recalculated the errors. The whole procedure is explained in the paper With that, we have a more clear error estimation.*

Can you describe the reasons for the differences in LF and MF fit based on their weighting of the data?
*The linear fit (LF) method applies a least-square approach, which means that every data point has the same distance to the best fit. The MF method by Clauset et al. (2009) is based on a maximum likelihood approach.*

It might be more interesting to consider if the fit is statistically significant.
*In the current version of the paper, we recalculated the exponent 100 times with a random selection of 70% of the measured lead widths. The mean over the 100 exponents then is the final result. The final exponent has a small standard deviation for both thresholds and methods. Thus, we argue that the fit is statistically significant.*

"Based on our current knowledge, previous studies (Table 2) focused on different regions in the Arctic and not on Antarctic regions." Here "Based on our current knowledge" is not needed in the sentence. *Revised.*

When comparing to other studies in the Arctic it should be clarified that you are not comparing like method for like. In particular the different sensors have different spatial resolution, which will constrain the range over which the power law fit is made.

*Yes, as suggested by the reviewers, we clarified the differences of the sensors and methods used in the literature.*

If you consider how there appears to be a decrease in exponent at smaller scales in your data, would this explain any of the differences to previous studies with lower resolution sensors?

*We did not discuss a scale dependency in our data, since our data got cut off due to the lack of wider leads. Additionally, we only have some measurements of wide leads, which could also explain a decrease in the exponent.*

**Answer to Anonymous Referee #1**

*We thank the editors Jenny Hutchings and Yevgeny Aksenov and the two anonymous referees for the helpful comments and their efforts reviewing the paper.*
*The major changes within the second manuscript are the following:*

- *We restructured the introduction, discussion and conclusions of the paper*
- *We included three different flow charts to explain the data selection and processing (Figure 1, Figure 2 and Figure 3) in Section 2 and 3. Therefore, we made changes within Figure 1, added Figure 2 and replaced Figure 3 with a completely new figure Section 3.1 was then rewritten and restructured to fit the flow charts*
- *We changed the names of the sea-ice types used for the surface-type classification according to the WMO Sea Ice Nomenclature*
- *We gave new names to the two used TOA-thresholds (open water (ON) and open water and nilas (OWN) threshold)*
- *We re-calculated the power-law exponents and their standard deviation (see Section 3.2.)*
- *We extended the discussion in regards to the difference between the values for the Arctic and in regards to our results for the Weddell Sea and their possible representation for the Antarctic sea ice*
- *We removed Section 4.3 and shifted it contents to Data (Section 2) and the discussion (Section 4)*

*Please find point-by-point answers to the comments below. The answers to the other referee and the editors note are separate comments.*

**Anonymous Referee #1**

The manuscript aims to compare lead-width distribution estimates from an Arctic set-ting to an Antarctic setting. Within the manuscript a set of lead-width distribution algorithm parameter estimates are being calculated using a set of Sentinel-2 images. The algorithm parameter was found to be lower compared to the estimates for the Arctic sea ice areas. As the Antarctic sea ice areas are significantly less investigated com-pared to the Arctic sea ice areas the subject is interesting and timely. Unfortunately, the manuscript is poorly written and it's very unclear what the overall aim and the novel scientific contribution is.

Specific comments

It would be beneficial if the method is also used on a set of images covering the Arctic sea ice to see if the change in parameter setting relates to the Antarctic conditions, the method or the satellite data. In the Abstract it is stated that you compare exponents from the Arctic sea ice that do not agree with each other to your estimates for the Antarctic sea ice. Why do the estimates from the Arctic sea ice not agree?
*We agree that it would be interesting to analyze images from the Arctic. However, such an analysis was not within the scope of this study. We will include this point in the discussion an elaborate more on the (regional) differences of the exponents.*
*The estimates from the Arctic sea ice do not agree to each other as discussed in the manuscript. It is unclear if this is related to the fitting method or the underlying data set. For future comparisons the same fitting method should be applied to estimate the exponents.*

As far as I can understand the nilas areas include both open water and nilas. Please confirm. It is unclear why these two lead types are not separated from each other, and if this separation was done in the scientific work by others that the lead parameter settings are compared to. This needs to be clarified.
*Yes, nilas is both open water **and** nilas. Therefore, we renamed the thresholds. We do not separate these from each other since the water in leads refreezes quickly depending on the surrounding temperatures.*
*But even when leads are covered with thin ice like nilas, they have properties that are characteristic for leads with regard to heat exchange, ice production and also being navigable by several surface vessels (see "WMO Sea Ice Nomenclature (WMO No.259, volume 1", 7.3 Lead,* https://library.wmo.int/index.php?lvl=notice_display&id=6772#.YAgyIsJ7mUk *(last access: 20.01.2021)*

Why are leads with dark-gray sea ice (up to 10cm thickness?) and light-gray (up to 30 cm thickness?) sea ice not included in the analysis? It is specifically stated in the methods section that you are investigating threshold for four different sea ice types. Dot he studies compared to within this paper include those other sea ice types?

*The surface classification are not connected to any measured thicknesses, since the Sentinel-2 Level-1C product only provides optical data, which is why the names correspond to the "grayness" of the ice.*
*For the calculation of the lead with distribution we decided to focus on leads covered by open water or open water and nilas only – mainly because previous studies also investigated leads with these surface types. This is necessary to make the obtained results more comparable. The other surface types could be used for different applications. We will clarify this in the corresponding sections.*

A schematic figure outlining the processing steps would aid the reader to understand the different steps within your method. How are the classified images validated? How many different individual leads were used in this study?

*In total, the number of leads is 2024 for the open water threshold and 3799 for the open water and nilas threshold.*
*Thank you for the recommendation to include a schematic figure. We changed three figures in the paper. Figure 1 describes the selection of the data, Figure 2 the creation of the Gaussian curves used for obtaining the thresholds and Figure 3 the measurement of the lead widths.*

Overall the manuscript is poorly focused, many sentences difficult to follow and references to appropriate work is missing. The presented aim is the lead-width distribution however, the manuscript focuses on the algorithm parameter setting and not on the outcome of the lead-width estimates. Please either change the aim of the manuscript or change the manuscript to reflect the aim.

*We will revise the manuscript according to your detailed suggestions below.*
*Regarding the focus, we restructured some parts of the paper. The ice type and threshold detection is only a prerequisite, which is necessary to identify leads. The main aim is then to use these leads to obtain a lead width distribution.*

Technical corrections

Why was the Weddell sea picked for this study? Does it have a high/low frequency of leads compared to other areas of the Antarctic sea ice areas? Work by e.g. Willmes and Heinemann (2016) have indicated that different parts of the Arctic Ocean have different regional characteristics.

*The Weddell Sea was chosen for its special scientific importance, its size, the size of the sea-ice cover during November to April and availability of cloud free images in the Southern Ocean. Furthermore, Sentinel-2 is a land mission and images over ocean are only available in the vicinity of land which restrict the regional selection. We included a discussion of the results in regards to the possible limitations occurring from the area selection. We included the Willmes and Heinemann (2016) paper for comparisons and further assessment of our results, thank you for suggesting.*

P1. R4-5. Unclear sentence please revise. *Changed*

P1. R7-8 after the comma is a repeat of what was said on row 4-5. Please remove. *Removed.*

P1. R12. Replace bigger with larger or greater than *Changed.*

P1 R21. Please insert a reference. *Reference added.*

P2 R34. How wide is a very narrow lead? *The "narrow lead" refers to the resolution of 10 m of our study.*

P3 R4-5. Unclear sentence please revise. *Section was restructured.*

P3 R6. Insert references to these studies. *Reference added.*

P3 R13-14. This is better suited in the discussion. *We agree and moved it.*

P3 R21. Are you using data from November to February or November to April? The way this is written now is confusing.
*The sentence is changed. The data is always part of the months between November to April, while being from the overall time period of November of 2016 to February of 2018.*

P4 R5. This section deals primarily with identifying thresholds and not classification of surface types. Consider changing the title. Moreover, the classification step is done in section 3.2 and should be reflected in the title there. I'm sorry but I could not follow how the lead width information was extracted from the images. Please clarify this.
*The section 3.1. is renamed "Threshold identification". We also restructured and rewrote parts of the whole method section, so that it is clearer which steps are part of the threshold identification and then the lead measurement (see Figure 3).*

P4 R9. Unclear sentence please revise. *Revised.*

P4 R16-18. Unclear sentence please revise. *Revised.*

P4 R28. Not only nilas but also open water. *Yes.*

P4. R30. What is apparent lead width? Please define.
*The "true lead width" would be measured if every lead width was measured orthogonal to the lead. Our measurement grid can not guarantee that, which means that we sometimes measure not the shortest width but the width of a line across the lead with an angle other than 90 degree. That is why we call the lead width "apparent lead width" similar to Wernecke and Kaleschke (2015) .*

Figure 2 Consider indicating the different TOA-reflectance areas that results in the different ice types, e.g. introduce a gray scale indicating classification areas. At the moment this figure is difficult to understand.
*We use Figure 2 to highlight how we estimated the used thresholds. A gray scale would probably shift the focus of the image.*

P5 Figure 3 text. What does "The swath does not cover the whole image area" mean? Either the Sentinel-2 image exist or not. Should image area be study area?
*The Sentinel-2 Level-1C products are always 100 km x 100 km products, but the swath of the satellite represented in the product does not always cover the whole area of the product, which is why the bottom right corner in the upper Band 4 image has a solid white triangle.*

P4 R30 -P5 R1. Unclear sentence please revise. *Revised.*

P6. R2. What is x_width? *x_width is the measured apparent lead width.*

P6 R3. Where did the values for parameter C come from? This needs to be clarified.
*The parameter C is the offset at the y-axis of the function and is therefore related to the number of measurements. We included more information.*

P6 R13-14. Please add the 10m to the equation, or specify how the 10m were added to the equation.
*The 10 m were added for the variable "step size", since the 10 m indicate the step size of the measured lead widths. We revised the sentence describing the equation 3.*

P6 R17-18. This section is not results but should be put in the method or data section.
*In Section 4.1 we use the thresholds determined using the method described in 3.1 and compare the resulting reflectances to albedo values from the literature. Thus, this is rather a discussion part - which we decided to combine with the Results section to improve the readability of the manuscript.*

P6 R30. References missing. *Reference added.*

P6 R29-30. Do the TOA values reported here also correspond to the values for the Arctic sea ice? The comparison here appears to be only for Antarctic data.
*As we stated above, this would be interesting to investigate in the future but is beyond the scope of this study.*

P7 R6. Compared to what? Figure 5 is difficult to see. As the discussion primarily revolve around the for the two different methods for the two different mediums, why not combine the open water areas in one figure and indicate the different methods in the legend? Or better yet combine everything into one figure for easier interpretation. Consider also changing the color scheme as the colors are too similar and it makes it difficult to analyze the figure.
*We disagree with putting everything (4 lines, two data sets) into one plot, since the lines then tend to overlap and the figure is crowded and it becomes difficult to understand the fine differences. We changed the colors of the figures, so that both figures use the same color for the same line.*
*We decided to create 2 figure panels for each of the two question of the problem. 1) The influence of the applied fitting method on the same data set, which is Figure 4. Figure 4 shows the combination of both methods only for the open water threshold (left figure) and only for the open water and nilas threshold (right figure). 2) The influence of the used threshold on slightly different data sets, where we can see that for example the influence of the data set for the ML method is really small. These nuances would be more difficult to highlight in one combined image.*

P8 R7-8. Unclear sentence please revise. *Revised.*

P9 R6-8. Unclear what you are trying to say here. Please discuss why different results are achieved and what this means.
*Here, "results" refers to the different power law exponents in the literature. We revised the sentence and extended the discussion of the differences of the results and what this can mean for future investigations.*

P9 R24-26. According to Table 2 the resolution of the Wadhams papers are 5m. It is unclear what you are saying here. Please revise.
*The Wadhams studies did not use satellites, but submarine missions, thus their possible spatial coverage is smaller. The advantages of satellite data are the increased spatial coverage and repeated overpasses. Here, we wanted to highlight the high spatial resolution of Sentinel-2 and the possible advantages compared to other satellite products.*

P9 R34-35. Please specify what these other data sources could be.
*Since Sentinel-2 only provides optical information a combination with field measurements for the validation of the surface types would be beneficial.*

Section 4.3. This section does not belong in the results but should be moved to the data section instead.
*We deleted section 4.3 and rearranged the content into the data section.*

Table 2. The table needs to be updated so that it is clear to the reader that the two values reported in the present study are for open water and nilas+open water areas. It is unclear from the present manuscript what the other reported values have used as criteria for their power law estimates? Are these values reflecting values for open water, nilas or leads generally? For the two different thresholds used in Marcq and Weiss (2012), do they also separate into two different lead types or are the two values a result of two different methods? If it is the latter please separate the values reported in the same way as is done with the present study results.
*We updated the table to make the differences between the results more obvious. Marcq and Weiss (2012) used two different thresholds for leads (with open water) based on two different luminescence thresholds.*

Reference
Willmes S. and Heinemann G, (2016), Sea-Ice Wintertime Lead Frequencies and Regional Characteristics in the Arctic, 2003–2015, Remote Sensing, 8(1),4,https://doi.org/10.3390/rs801

**Answer to Anonymous Referee #2**

*We thank the editors Jenny Hutchings and Yevgeny Aksenov and the two anonymous referees for the helpful comments and their efforts reviewing the paper.*
*The major changes within the second manuscript are the following:*

- *We restructured the introduction, discussion and conclusions of the paper*
- *We included three different flow charts to explain the data selection and processing (Figure 1, Figure 2 and Figure 3) in Section 2 and 3. Therefore, we made changes within Figure 1, added Figure 2 and replaced Figure 3 with a completely new figure Section 3.1 was then rewritten and restructured to fit the flow charts*
- *We changed the names of the sea-ice types used for the surface-type classification according to the WMO Sea Ice Nomenclature*
- *We gave new names to the two used TOA-thresholds (open water (ON) and open water and nilas (OWN) threshold)*
- *We re-calculated the power-law exponents and their standard deviation (see Section 3.2.)*
- *We extended the discussion in regards to the difference between the values for the Arctic and in regards to our results for the Weddell Sea and their possible representation for the Antarctic sea ice*
- *We removed Section 4.3 and shifted it contents to Data (Section 2) and the discussion (Section 4)*

*Please find point-by-point answers to the comments below. The answers to the other referee and the editors note are separate comments.*

**Anonymous Referee #2**

While the study provides interesting statistics about leads in the Weddell Sea, it does not offer details about the uncertainties implied in the data or the methodology (i.e. setting the thresholds). The subject is certainly timely and will have immediate implications, given the impacts of climate change on the polar regions, but I think with such as short manuscript it suits more a letter-style paper. I am not sure if TC allows this.
*By including the suggestions by the referees, the paper became more detailed and a bit longer than before. Thus, we want to continue to handle the manuscript as a full paper and not change it into a letter.*

The manuscript is fairly well written though I found some sentence structure can be improved, e.g. "we noticed that on products with wide leads ...." or "The goal of the classification is to get thresholds...". A better word would be "to identify thresholds". The "amount of lead" should be "the number of leads", and so on. I am not including more details but the authors and the site editor(s) can adjust the style.
*Thank you, we checked the paper again and rephrased several sentences.*

MAJOR ISSUES:

The title suggests that results of lead distribution apply to the Antarctic ice cover. I don't think this would true. The authors concentrate on the Weddell Sea as a case study. There are two features in this region that make it unique, the dominant old ice and the gyre (Weddell Sea gyre). That should make the leads in this area unique. Perhaps a better title should specify Weddell Sea only.
*The title should imply that this is the first lead-width distribution for an area (the Weddell Sea) of the Antarctic sea ice, therefore the label case study. We included more information on the study area.*

The authors admit that in presence of snow, the reflectance measurement will not an indicator of the underlying ice cover. I don't see how they addressed this problem. Please clarify.
*We assume that nilas has no or only a really thin snow cover, since snow would brighten the appearance of the surface. Additionally, snow has an insulating effect and therefore inhibits the heat exchange for an example which would reduce the "lead effect" of an area where nilas is covered with snow.*

*This is also the reason, why the names of our surface categories do not indicate any age of the ice or thickness of the ice, since – as you stated – the ice in the category "sea ice covered with snow" can be really different. But since we use the thresholds from the classification only for measuring lead widths, we are confident in the thresholds.*

While leads are mostly covered with open water or thin ice, they also serve as a path for broken thick ice. The authors set thresholds on reflectance to identify the surface of the lead, OW, nilas, grey ice, etc. If I understand it correctly, the authors calculate the lead statistics for each type separately (Fig. 5 shows OW and nilas), but a lead may have all types of ice plus OW. What if the lead in your data is composed of the five ice types you referred to? Would that be included in the lead statistics? The answer is still not clear to me after reading the entire manuscript.

*We only investigate leads with open water or leads with nilas and open water (OW and OWN threshold in the current version of the paper). The other surface categories are not used for the lead-width estimation. Also, the nilas threshold contains areas covered with nilas and open water. This then covers for example also the case of a lead, where one area starts to refreeze while the other part is still open. Thus, in that case we measure the whole lead width. We decided to omit leads that are covered by brighter (thicker) ice, since this has been common practice in many other studies (e.g. Marcq and Weiss (2016) and Lüpkes et al. (2008)) and thus makes our results more comparable.*

*Lüpkes, Christof, et al. "Influence of leads in sea ice on the temperature of the atmospheric boundary layer during polar night." Geophysical Research Letters 35.3 (2008).*

The authors use Reflectance to set thresholds to identify surface types. However, Reflectance is angular-dependent measurement, therefore it is not a property of the surface. Albedo is a property of the surface because it is the integration of the angular reflectance. The comparison between albedo and reflectance on page 6 lines 24-26 cannot be used to confirm the reflectance thresholds. To proceeds with the reflectance in this study, all leads must be viewed with the same angle from the different satellite orbits.

*This is an appropriate simplification because it is not practical to restrict the observed reflection angle. The reflectance anisotropy could in principle be estimated using many observations over rather static scenes (e.g. Roy et al, 2017). However, we do not expect a significant influence of a potential anisotropy on the resulting lead with distribution.*

*Roy, David P., et al. "Examination of Sentinel-2A multi-spectral instrument (MSI) reflectance anisotropy and the suitability of a general method to normalize MSI reflectance to nadir BRDF adjusted reflectance." Remote Sensing of Environment 199 (2017): 25-38.*

MINOR ISSUES:

INTRODUCTION: There are many more references that can be used to review the work on lead width geometry and width distributions. The authors should check and quote more than the 2 references used here. Moreover, the Introduction jumps too much between different themes. I would recommend the authors to re-write while grouping themes in separate paragraphs.
*Thank you for the suggestion, we rewrote the introduction.*

P 1 L18: leads may not always form in linear shape. *We replaced "thus" by "often" to also include other possible shapes.*

P2 L1: what is "coupled climate models"? *"coupled climate models" commonly refers to models consisting at least of an ocean or land and an atmosphere component which are coupled to each other.*

P2 L15: "Reiser et al. (2020) introduced a retrieval algorithm for lead fraction in the Antarctic, but these studies indicate that knowing about lead-width distributions is beneficial when estimating heat transfer". It is not clear how this can be a conclusion from a lead identification study. For one thing, it can only be a conclusion from a heat flux study but more importantly is a logical conclusion. It does not need a study.
*The mention of Reiser et al. (2020) was supposed to highlight that there are lead-fraction products available, also for the Southern Ocean, but that the lead width is still missing. And that other studies (the ones named before) have indicated, that the lead width is important.*

P2 L28: "To narrow down the effect of the fitting methods, we applied ...". Please change the sentence as the purpose of applying the two methods is not clear. *Changed.*

P3 L4:"The determined thresholds for leads covered with open water .." Threshold on what? Reflectance? *Yes, the threshold is a TOA reflectance threshold.*

P3 L5: the sentence "Two threshold are applied to identify .." is not understood. Please rephrase. *Revised.*

Figure 1: I find some of marked frames of the satellite images odd (with 3 or 5 sides instead of 4). Why is that? Also, in the caption, instead of "with shelf ice" it is better to use "including shelf ice border measured with . . . " then specify the satellite radar sensor
*The form of the products is due to the differences of the satellite swath and the processing grid of the ESA which results in areas without a satellite measurement in the image (as explained and shown in Figure 3 with the missing corner in the bottom right corner in the upper image). We decided to use the "real image outlines" for the map to show the product size.*

DATA:

P3 L10: attach the spatial resolution to the band. *The resolution of each band is not directly indicated, because it is irregularly distributed, and we only use the 10 m resolution of Band 4. See:* https://sentinel.esa.int/web/sentinel/user-guides/sentinel-2-msi/resolutions/spatial

P3 L11: please include the spectral band for channel 4 (665 nm). *Included.*

Also, in Line 12, provide a reference to support your statement - that it is best for thin detection. I am not sure about the reason. Please specify. *We manually compared Band 2 – 4 and 8 with each other in areas, where we expected thin ice in leads. After visual inspection we and decided to use Band 4. We included the statement in the text.*

P3 L13: which Landsat? When I read this statement, I expect to see use of the coincident SPOT and LANDSAT data to support your finding. But they are mentioned here only to refer to their future use. Using these data will add value to the manuscript.
*This information is a citation from Drusch et al. (2012), but at least Landsat 8 crosses the Equator every day at 10 a.m., while Sentinel-2 crosses the Equator at 10:30 am.*
*The scope of the study was only to use Sentinel-2, this statement is just an outlook and is moved to the end of the paper – according to the suggestion by reviewer #1.*

Table A1: why A1? It is not in an appendix? I would recommend inserting in the main text at the end of Section 2. *We decided to put the table in the appendix due to its size, but we changed it for the current version.*

METHODS:

The use of n=2 and n-3 in Equation 1 for light-grey ice and dark-grey ice is not clear to me. I see clearly the use of n=1 for the other types in Figure 2. I don't think there is something wrong here. It is just my failure to understand. Please re-phrase.
*The n indicates the number of Gaussian curves which were combined to one function for fitting the histograms. By using n>1 we can account for multiple maxima within a distribution. We rephrased the explanation of n.*

Also, according to my information, what you call dark-grey and light-grey ice, should be grey ice (10-15 cm thick) and grey-white ice (15-30 cm thick). These are nomenclature set by WMO and used in operational ice centers. You can refer to the document that defines ice types, published by the Canadian Ice service: MANICE (2005), "Manual of Standard Procedures for Observing and Reporting Ice Conditions", Canadian Ice Service – Environment Canada, ISBN 0-660-62858-9, Catalogue No. EN56-175/2005.
*Since we only have information about the brightness of the ice and not of the thickness, we created our own terminology to indicate the brightness of the ice. However, we likes your suggestion of using the WMO standard nomenclature and renamed our categories.*

Figure 3: in the caption "The upper border of every image is 50 km wide". This is not the swath. Please clarify what you mean by upper border.
*We mean that the width of the figure corresponds to 50 km.*

This figure shows the OW in the lead. What about other ice cover, nilas, grey ice, etc.?
*Figure 3 is used to illustrate the lead detection algorithm described in Section 3.2. The figure on the right shows the leads for the open water and nilas (OWN) threshold. The other surface types are not used for lead-width measurements, which is why we do not show them to not confuse the readers.*

RESULTS

P6 L21: the sentence "the TOA reflectance is only measured passively in the absence of clouds …" is not clear. Please rephrase. *Rephrased.*

P7 L5: Nilas threshold cover also open water? Why? There is overlap between nilas and OW, but the threshold does not cover both.
*The nilas threshold covers both, which is why we renamed it to open water and nilas (OWN) threshold.*

P7 L6-21: it would be better to expand on the difference between lead statistics in the Arctic and Antarctic in a separate sub-section. Most of the lead studies in the Arctic are conducted in the Beaufort Sea area. Some organized presentation should show the differences in the results, with related reasons if possible, and then comparison with findings from the Weddell Sea area.
*We restructured the discussion and added more in regards to the regional influences.*

CONCLUSION

*Before you mention about the method, you should mention the data used in the study. Ok*

---

## Author Response (AR3)

**Answer to Anonymous Referee #1**

*We thank the anonymous referee for the additional comments on the second version of our paper. Please fine the point-by-point answers to the comments below.*

**Technical corrections**
Throughout the manuscript esa -> ESA *Changed.*

Some of the figures are using a) and b) and some Left: Right:. Please update this for consistency. *Changed.*

P1 R3-4. Unclear sentence after the comma please revise. *Revised.*

Fig. 1. Satellite Radar -> satellite radar *Changed.*

P4. R3. What is suitable? Please define. *Suitable is replaced with "Large enough".*

Section 2. The order in which the information is presented is a bit back and forth, the 20 images are selected (P3 R17) before the description on the criteria for the selection, e.g., cloud free and time of year, is presented on P4 R1-7. Please go through this section to ensure a logical continuation of the text. How many images were excluded as they didn't fulfil the set criteria? *Mainly deleted the first sentence of the paragraph. Sadly, we did not keep record of the total number of excluded images, which is why we did not include the number.*

P5. R8. First two should be to. *Thanks!*
The first two paragraphs on Page 3 section 3.2 could ideally be integrated as it is now there is a lot of backtracking. *We assume you mean page 6 and removed the first introductory sentence/paragraph. From our point of view, the paragraph about the "apparent lead width" should stay there.*

P5 R14. Image -> images? *"On every band 4 image" is supposed to be singular.*

Figure 4 is referenced to before Figure 3 in the manuscript. Please correct. Many figures appear in the text before the actual figure is shown, sometimes this is a full page apart. Please correct. *Changed the order of Figure 3 and 4. The final placement of the Figures within the text is up to the typesetting.*

P5 R18-19. The terminology used in part of the methods section doesn't live up to the standard one would expect from TC. Based on section 3.1 I would say that you use a supervised Bayesian classifier with a non-parametric PDF and estimate the PDF with kernel density estimation with Gaussian kernel (also known as Parzen windows). A suggestion is to use these terms when describing the method as it'll be easier to understand for researchers used to working with pixel-wise classification. *We did not use the method described by the reviewer, which is why we described it in the method section as we did.*

P6 R7-9. Open water may refreeze quickly, but this depends on the surrounding temperatures. Please update the sentence. *The sentence already mentions the surrounding temperatures.*

Fig. 4. The error areas indicated in this figure are not correct. What is indicated is the pixels that are nilas but are classified as open water, but not the ones that are open water that gets classified as nilas. Similarly for the nilas vs gray sea ice, and gray sea ice vs. gray-white seas ice, and gray-white sea ice vs sea ice covered with snow. Please update the figure with a complete set of errors. Moreover, there should also be a lower threshold for the OW class, as the values below approx. 0.05 are only nilas and not OW. *The error shows the overlap of two TOA-reflectance distributions, which occurs below the set threshold. Thus, it shows where we manually classified pixels with the same TOA-reflectance in different sea-ice surface categories. We then only account for the overlap error below the set threshold, which is why it looks like "half of the error".*

How were the error estimates presented in Table 2 derived, do they include the total error or only half of the errors as indicated in Figure 4? *They include the overlap error shown in Figure 4. To clarify that we renamed "error" to "overlap error" in Figure 4 (now Figure 3).*

P9 R 14. Comparably -> comparatively *Changed.*

P9R14. What is this sentence referring too? The separation between open water and nilas? *No. The differentiation between open water and nilas was more easy than between the other sea-ice surface types.*

P9R16. …thresholds (0.10 for OW and 0.16 for OWN) were used for the lead detection and agree with values from previous… *Changed.*

P10 R6-8. Do you mean that you measured leads with a width from 10m and upwards? *Yes.*

P10 R9. What do you mean with feature? Please clarify. *The feature is that there are less measurements as possibly expected.*

P11 R2-3. Why would leads with OWN be wider? *OWN also includes wider leads, since the edge of the lead starts to freeze and contains nilas, while the middle part is only covered with open water.*

P13 R12. Which Landsat? *Landsat 8*